# *WebFactory*: Automated Compression of Foundational Language Intelligence into Grounded Web Agents

**Sicheng Fan**[1,2]**, Qingyun Shi**[1,2]**, Shengze Xu**[3]**, Shengbo Cai**[2,4]**,**
**Tieyong Zeng**[3]**, Li Ling**[1]**, Yanyi Shang**[2]**, Dehan Kong**[2]

[1]Fudan University    [2]IMean AI
[3]The Chinese University of Hong Kong    [4]Tsinghua University

## Abstract

Current paradigms for training GUI agents are fundamentally limited by a reliance on either unsafe, non-reproducible live web interactions or costly, scarce human-crafted data and environments. We argue this focus on data volume overlooks a more critical factor: the efficiency of compressing a large language model's (LLM) latent knowledge into actionable agent behavior. We introduce *WebFactory*, a novel, fully automated closed-loop reinforcement learning pipeline for GUI agents, systematically compressing LLM-encoded internet intelligence into efficient, grounded actions. Our pipeline features a process of *scalable environment synthesis → knowledge-aware task generation → LLM-powered trajectory collection → decomposed reward RL training → systematic agent evaluation*. Remarkably, our agent demonstrates exceptional data efficiency and generalization. Trained on synthetic data from only 10 websites within *WebFactory*, it achieves performance comparable to GUI agents trained on same amount of human-annotated data from a much larger set of environments. This superior performance is consistent across our internal offline and online transferring benchmarks, where our agent also significantly outperforms the base foundation model. We further provide critical insights into the "embodiment potential" of different LLM foundations, offering a new axis for model evaluation. This work presents a scalable and cost-effective paradigm for transforming passive internet knowledge into active, grounded intelligence, marking a critical step towards general-purpose interactive agents.

## 1 Introduction

The advent of Large Language Models (LLMs) has marked a paradigm shift, creating what we term "internet-scale intelligence"—the rich world model and reasoning capabilities compressed from the vast internet corpus (Ouyang et al., 2022). Yet, this intelligence remains descriptive, not actionable. While an LLM's knowledge represents a powerful compression of digital experience, an embodied agent's single action in a GUI—a click or keystroke—is an exponentially deeper compression, translating abstract intent into tangible environmental change. Bridging this fundamental "semantic-to-action gap" is the central challenge in creating capable GUI agents; LLMs *know* about GUI interactions, they lack the *grounding* to reliably *perform* them in complex and dynamic GUI environments (Shi et al., 2017; Liu et al., 2018; Chezelles et al., 2024).

Current attempts to bridge this gap are caught in a dilemma between scalability and control. On one hand, reliance on human labor presents a two-fold bottleneck: beyond the immense cost and inherent biases of annotating thousands of trajectories (Deng et al., 2023; Luo et al., 2025), the painstaking, manual synthesis of high-fidelity environments can itself consume weeks of expert effort. On the other hand, training on the live web offers scale but sacrifices control; it is a chaotic environment where non-determinism, safety risks, and noise present formidable barriers to reproducible research (Zhou et al., 2024; Miyai et al., 2025; Garg et al., 2025). As a result, neither approach offers a sustainable path toward creating truly scalable and robust agents.

To overcome these limitations, we argue for a paradigm shift: instead of treating LLMs as mere components to be fine-tuned, we can leverage them as the architects of their own embodiment. We introduce the concept of **Intelligence Compression Factory**: a closed-loop, end-to-end pipeline that systematically transforms the descriptive, internet-scale intelligence of LLMs into grounded, actionable behavior. As shown in Figure 1, this factory operates not on the noisy live web(Pan et al., 2024), but within a high-fidelity, fully observable offline environment. Replicating real-world websites in this manner eliminates non-determinism and safety concerns, thereby creating the ideal conditions for our factory to operate. Here, an LLM-driven, knowledge-aware task synthesizer can generate a virtually infinite stream of diverse and executable tasks, shattering the bottleneck of human annotation.

Our key contributions are the design, implementation, and validation of this factory:

- **High-Fidelity Offline Web Environment:** An open-source and reproducible suite that faithfully replicates production websites, providing strict controllability and full observability while eliminating noise, privacy concerns, and non-determinism inherent to live web interaction.

- **Knowledge-Driven Task Generation:** A mechanism that leverages environment observability and LLM knowledge to automatically synthesize diverse, executable, and unbiased task instructions with unambiguous ground-truth answers, removing reliance on costly human annotation.

- **Scalable Trajectory Generation:** Integration of strong LLM executors (e.g., OpenAI's `computer-use-preview`) within the controlled environment to generate large-scale, high-quality interaction trajectories. A filtering process ensures reproducibility and correctness, while a novel "behavioral intent alignment feedback" further enhances information retrieval tasks.

- **Reinforcement Learning with Unified Action Space and Decomposed Reward:** An RL training framework supporting GRPO and related algorithms. We design a unified action space and a decomposed reward function that combines structural format validation with fine-grained accuracy (action type, click point, input text). For retrieval tasks, normalized F1-based scoring stabilizes optimization and improves robustness (Christiano et al., 2017; Ouyang et al., 2022).

- **Robust Evaluation Protocols:** Comprehensive evaluation at both the task level (via key-node tracking) and sub-task level (via grounding metrics), enabling systematic and reproducible assessment of agent capabilities.

- **Open-Sourced Toolchain:** A fully released, extensible toolkit including environments, task generators, training pipeline, and evaluation tools, supporting scalable and reproducible research across diverse web domains.

Agents trained in *WebFactory* exhibit superior performance and data efficiency. On our internal offline and online benchmarks, they consistently outperform both the base foundation model and existing agents trained on equivalent volumes of human-annotated data(Luo et al., 2025). More strikingly, despite being trained on only 10 websites, our agent achieves competitive performance on general benchmarks against counterparts trained on a much broader corpus of human data. This success offers compelling evidence for the "intelligence compression" philosophy.

Beyond empirical performance, we introduce the concept of **"LLM embodiment"** , quantifying how effective foundation LLM tokens are transformed into grounded agent intelligence. Our analysis reveals that different foundation models possess a varying potential for embodiment, offering a new axis for model evaluation. Our findings also highlight the critical roles of full environment observability in maximizing training efficacy.

In summary, this work presents a scalable, safe, and cost-effective approach to transforming LLMs' descriptive intelligence into actionable GUI agent behaviors. We further propose that the agent scaling law should be refined beyond data volume to account for a model's efficiency in intelligence compression and its inherent capability for embodiment. While validated here in GUI settings, this paradigm holds strong promise for more complex physical embodied environments (Chevalier-Boisvert et al., 2018; Shridhar et al., 2020).

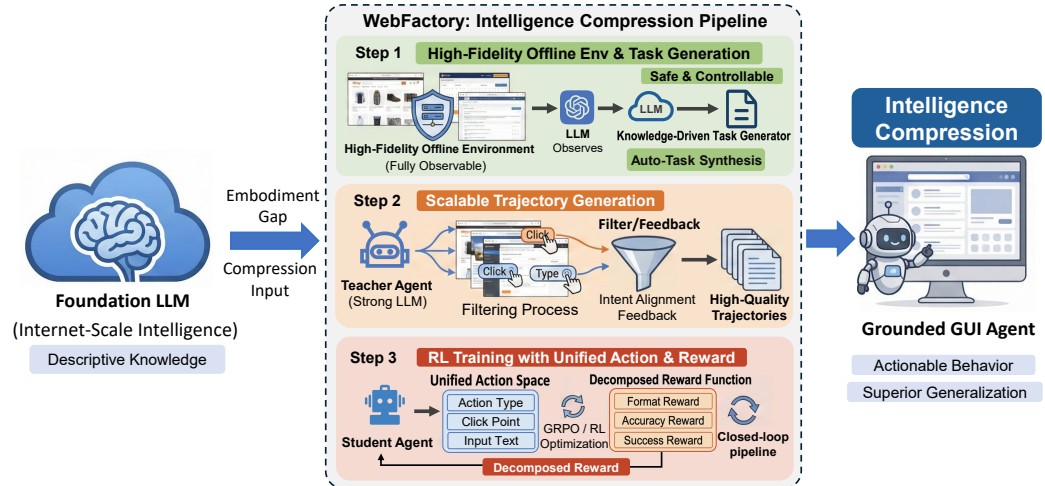

Figure 1: **Overview of the *WebFactory***, which compresses foundation-model intelligence into grounded GUI agents through three stages: high-fidelity offline environment & task synthesis, scalable trajectory generation, and unified-action RL training.

## 2 METHOD

### 2.1 A HIGH-FIDELITY, FULLY CONTROLLABLE WEB ENVIRONMENT

To enable scalable data generation and automated RL training for web agents, we develop a fully controllable offline environment that preserves the structural richness of production sites while guaranteeing strict reproducibility. A central component is our LLM-assisted synthesis pipeline, which automatically generates realistic websites—including layouts, workflows, and content—enabling low-cost, rapid expansion of training domains without manual engineering. This design achieves three critical objectives: (i) cost-effective synthesis of large-scale, high-quality training data, (ii) safe and systematic RL experimentation without real-world consequences, and (iii) stable, versioned benchmarks for reproducible evaluation.

The environment eliminates common deployment obstacles: sites boot into pre-authenticated sessions with seeded profiles, bypassing login/MFA requirements; anti-automation defenses (CAPTCHA, bot detection) are disabled to isolate agent capabilities; and all content is versioned in static datasets (e.g., 'Data.js') for exact reproducibility. Full access to frontend code, databases, and interaction logic facilitates rapid iteration and instrumentation.

We curate ten site families spanning key web activities: e-commerce, information search, travel planning, employment, communication, and enterprise services. These sites feature diverse UI patterns—from simple forms to drag-and-drop interfaces and hover-triggered menus—providing comprehensive coverage of web interaction paradigms.

The entire codebase is open-source, enabling researchers to extend the site collection or implement custom tasks. Task difficulty is adjustable across data complexity (catalog size, network density), UI complexity (multi-level navigation, drag-and-drop, hover menus), and workflow depth (from simple lookups to multi-step executions). This flexibility supports targeted evaluation of key competencies: information retrieval, form completion, navigation efficiency, and constraint-based decision-making.

**Pipeline integration.** The environment supports the end-to-end training pipeline described in Sec. 2.2. It exposes ground-truth data and site knowledge for task synthesis, enforces trajectory correctness during generation, and enables automatic reward computation for RL. For information-retrieval tasks, canonical answers are directly accessible from the data layer. This infrastructure serves both as a data-generation platform and as a versioned benchmark for reproducible evaluation.

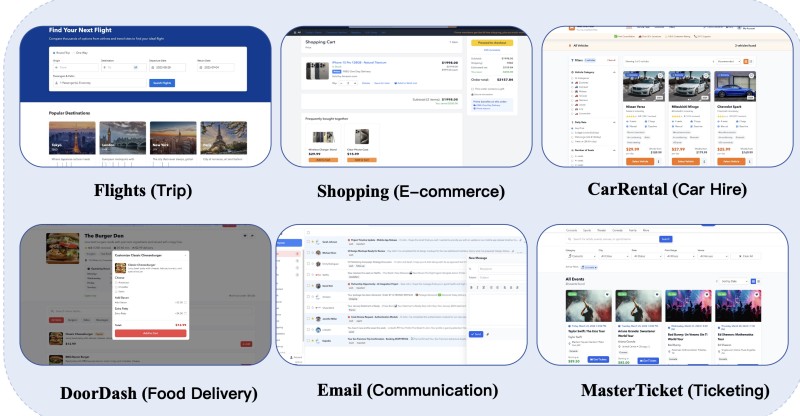

Figure 2: Representative offline websites from our curated environment (6 of 10 shown).

## 2.2 A KNOWLEDGE-DRIVEN RL TRAINING PIPELINE FOR WEB AGENTS

### 2.2.1 KNOWLEDGE PRESERVATION & TASK GENERATION

**Knowledge-driven task generation.** A critical advantage of our fully observable environment is the ability to guarantee task validity and answerability. For each site, we extract a machine-readable knowledge specification capturing: (i) the navigation graph with permissible page transitions, (ii) page-level semantics and affordances, and (iii) canonical interaction flows (e.g., browse $\rightarrow$ detail $\rightarrow$ cart). This complete observability eliminates common data generation pitfalls—tasks referencing non-existent pages, unavailable information, or infeasible actions are prevented by design.

Leveraging this knowledge, we generate two complementary task families:

(a) *Operation tasks* evaluate long-horizon interaction competence through state-changing actions (e.g., "Add iPhone 17 with 256GB storage to cart"). These are synthesized by traversing the navigation graph to ensure all generated procedures are executable on the actual site.

(b) *Information-retrieval tasks* pose queries with guaranteed answers drawn directly from the observable data layer (e.g., "What are Cafe A's weekend hours?"). Since all site data is accessible, we verify answer availability before task generation and compute the exact navigation path required for retrieval. This approach produces unambiguous ground-truth answers essential for both supervised learning and automated reward computation (see Listing 1 for an example).

The full observability thus transforms traditionally unreliable task generation into a deterministic process, ensuring every synthesized task is both executable and verifiable—a prerequisite for scalable training and evaluation.

### 2.2.2 BATCH DATA GENERATION AT SCALE

Given a predefined task set, we use a strong executor (OpenAI's computer-use-preview) within this offline environment to execute tasks and collect trajectory data. A filtering pipeline removes low-quality traces via (i) state-replay checks, (ii) key-node coverage, and (iii) answer validation for retrieval tasks. In addition, site-exposed auxiliary knowledge assists the executor and enables extra consistency checks, improving both accuracy and yield. Together, these properties make large-scale trajectory generation routine and low-cost while preserving reproducibility: the result is a scalable corpus of high-quality data suitable for SFT, offline RL, or hybrid training. Appendix F provides a detailed description of Trajectory Dataset Statistics & Distributions.

### 2.2.3 REINFORCEMENT LEARNING FROM GENERATED TRAJECTORIES

We build upon the GUI-R1 framework (Luo et al., 2025) and extend it to support information retrieval tasks in web environments. While the original framework focuses on action-oriented

```
{
  "id": "task_retrieval_017",
  "site": "MealDash",
  "start_url": "/mealdash",
  "goal": "Search for Cafe A, open its detail page, and tell me the Sunday opening
  ↪  time, formatted as HH:MM in 24-hour style.",
  "expected_answers": [
    "11:00",
    "11 am",
    "opens at 11:00"
  ],
  "key_nodes": [
    "search_box",
    "results_list",
    "cafe_detail_page"
  ]
}
```

Listing 1: Example schema for a retrieval task

GUI manipulation, we adapt it to handle data acquisition tasks by introducing a specialized `get_final_answer` action and corresponding reward mechanisms for answer evaluation.

We optimize a policy for web-based GUI agents operating in a structured action space. Each action at step t is a tuple

$$\mathbf{a}_t = \{a_t^{\text{act}}, a_t^{\text{point}}, a_t^{\text{text}}\}, \tag{1}$$

where $a_t^{\text{act}} \in \{\texttt{click}, \texttt{double\_click}, \texttt{type}, \texttt{scroll}, \texttt{keypress}, \texttt{drag}, \texttt{get\_final\_answer}\}$ denotes the action type, $a_t^{\text{point}} = [x, y]$ (or $[[x_1, y_1], [x_2, y_2]]$ for $\texttt{drag}$), and $a_t^{\text{text}}$ contains input text or directional parameters (e.g., UP/DOWN for $\texttt{scroll}$). Generated trajectories populate a replay buffer $(s_t, \mathbf{a}_t, R_t, s_{t+1})$.

**Reward.** Let $R_f$ be the format reward and $R_{\text{accuracy}} \in [0, 1]$ be the task-specific accuracy reward. The per-step reward is

$$R_t = \alpha R_f + \beta R_{\text{accuracy}}, \tag{2}$$

where $\alpha, \beta$ are weighting coefficients.

**Accuracy Reward.** We employ hierarchical validation: action type must match before evaluating action-specific parameters. Let $\mathcal{A} = \{\texttt{click}, \texttt{type}, \texttt{scroll}, \texttt{drag}, \texttt{get\_answer}, ...\}$ be the action set. The accuracy reward is:

$$R_{\text{acc}} = \begin{cases} 0, & \text{if } a^{\text{type}} \neq gt^{\text{type}} \\ \mathbb{I}[a^{\text{coord}} \in gt^{\text{bbox}}], & \text{if } a^{\text{type}} \in \{\texttt{click}\} \\ \mathbb{I}[F_1(a^{\text{text}}, gt^{\text{text}}) \geq \tau], & \text{if } a^{\text{type}} \in \{\texttt{type}, \texttt{scroll}\} \\ \max_{r \in \mathcal{R}} \mathbb{I}[F_1(a^{\text{text}}, r) \geq \tau], & \text{if } a^{\text{type}} = \texttt{get\_answer} \\ \mathbb{I}[\|a^{\text{drag}} - gt^{\text{drag}}\|_2 \leq \epsilon], & \text{if } a^{\text{type}} = \texttt{drag} \\ 1, & \text{otherwise} \end{cases} \tag{3}$$

where $\tau = 0.5$ is the F1 threshold, $\epsilon$ is the drag tolerance, and $\mathcal{R} = \{r_1, ..., r_K\}$ contains equivalent answers for retrieval tasks. Text comparison uses normalization $\text{norm}(\cdot)$ for case/punctuation/format invariance.

**Format reward.** $R_f$ validates the structural integrity: proper JSON formatting, valid action types from the web action set, appropriate parameter types, and conditional requirements (e.g., text required for $\texttt{type}$ actions, directional strings for $\texttt{scroll}$).

### 2.2.4 CLOSED-LOOP PIPELINE

We integrate the controllable environment (Sec. 2.1), knowledge & task generation (Sec. 2.2.1), large-scale trajectory collection (Sec. 2.2.2), and RL training (Sec. 2.2.3) into an open, fully scriptable pipeline that operates with minimal human oversight.

The pipeline proceeds as follows: (1) **Knowledge & data materialization:** for each site, construct a knowledge pack comprising the navigation graph, page semantics and affordances, canonical flows, and an explicit data snapshot for downstream use; (2) **Task synthesis:** combine template- and LLM-based generation, with automatic validators (schema, visibility, reachability) to produce the task set $\mathcal{T}$; (3) **Trajectory generation and filtering:** execute $\mathcal{T}$ with a strong agent in the offline suite, enforcing deterministic replay, key-node coverage, and answer checks to yield the replay buffer $\mathcal{B}$; (4) **RL training:** optimize $\pi_\theta$ in the unified action space to maximize $J(\theta)$ using a decomposed reward that combines format validation with fine-grained accuracy (action type, click location, input text), with retrieval answers scored by normalized $F_1$; and (5) **Evaluation:** scripted replays with key-node–aligned process metrics and normalized answer matching, eliminating the need for human raters.

## 3 EXPERIMENTS

We conduct comprehensive experiments to validate the effectiveness of our knowledge-driven reinforcement learning pipeline for web agents. Our evaluation spans three key dimensions: (1) testing the synergy of knowledge- and data-driven approaches, (2) benchmarking trained agents across multiple evaluation suites, and (3) analyzing performance when instantiated with different foundation models.

### 3.1 EXPERIMENTAL SETUP

#### 3.1.1 DATASETS AND BENCHMARKS

We consider three levels of benchmarks. **Offline Website Benchmark:** an internal benchmark with 100 tasks across 10 offline websites, covering both operational tasks (e.g., adding items to a cart) and information-retrieval tasks (e.g., extracting product specifications). Tasks are grouped into three difficulty levels: simple (single-step), medium (3–5 steps), and complex (>5 steps). **Offline-to-Online Transfer:** to measure generalization, we test on three representative online platforms—Amazon, Airbnb, and Booking—with 30 tasks per site, evaluating transfer from controlled offline training to real-world execution. **Public Benchmarks:** we further assess generalization on GUI-Act-Web (Chen et al., 2024), OmniAct-Desktop (Kapoor et al., 2024), and GUI-Odyssey (Lu et al., 2024), which provide standardized tasks for web and GUI agents.

#### 3.1.2 EVALUATION METRICS

We report three metrics. Task Completion Rate (TCR) measures the percentage of successfully completed tasks. Action Accuracy is decomposed into action-type accuracy (Type), grounding accuracy (GR), and success rate (SR). Step Efficiency measures the ratio of executed steps to optimal path length.

#### 3.1.3 BASELINE MODELS

We compare against three representative baselines. QwenVL2.5-3B is an untuned vision–language foundation model (Bai et al., 2025). GPT-4o is OpenAI's multimodal model with strong zero-shot capability (Achiam et al., 2023). GUI-R1-3B is a web agent trained with large-scale human-annotated data (Luo et al., 2025).

Table 1: Task generation quality under different config

| Config | Exe. (%) | Val. (%) | Div. | Cmplx. (%) |
|---|---|---|---|---|
| No Knowledge/Data | 31.3 | 42.3 | 0.31 | 8.2 |
| Data-Only | 56.3 | 68.7 | 0.52 | 15.6 |
| Knowledge-Only | 62.5 | 71.2 | 0.64 | 22.3 |
| Knowledge + Data | **86.3** | **92.6** | **0.84** | **35.7** |

Table 2: Trajectory data quality

| Metric | No-Kn. | Kn. |
|---|---|---|
| SR (%) | 42.6 | **84.3** |
| Steps | 15.7 | **9.8** |
| VD (%) | 58.3 | **89.6** |

Table 3: Performance on the internal offline website benchmark for operational tasks and information retrieval, reported by task completion rate (TCR), efficiency, accuracy, and F1 score.

| Model | Operational Tasks | | | Information Retrieval | | |
|---|---|---|---|---|---|---|
| | TCR(%) | Efficiency | Acc.(%) | TCR(%) | F1 Score | Acc.(%) |
| QwenVL2.5-3B | 18.3 | 0.32 | 41.2 | 15.7 | 0.28 | 36.4 |
| GPT-4o | 26.7 | 0.41 | 48.6 | 22.3 | 0.35 | 42.8 |
| GUI-R1-3B | 68.2 | 0.78 | 85.3 | 64.6 | 0.76 | 81.2 |
| WebFactory-3B | **71.8** | **0.82** | **87.6** | **67.3** | **0.79** | **83.4** |

## 3.2 EFFECTIVENESS OF KNOWLEDGE AND DATA-DRIVEN APPROACH

### 3.2.1 IMPACT ON TASK GENERATION QUALITY

We first validate how knowledge and data-driven methods improve task generation quality. For each configuration, we generate 80 tasks and evaluate their executability on actual websites. Table 1 presents the results under different configurations.

The combination of knowledge and data substantially improves task executability from 31.3% to 86.3%. Task validity increases from 42.3% to 92.6%, while knowledge-driven methods leverage website structure information to generate more diverse and complex multi-step interaction tasks, increasing complex task proportion by 4.4× compared to the baseline.

### 3.2.2 IMPACT ON TRAJECTORY DATA QUALITY

Knowledge-driven methods substantially improve trajectory generation quality. As shown in Table 2, the success rate nearly doubles (42.6% → 84.3%), while the average number of steps decreases by 38% (15.7 → 9.8), indicating more efficient task execution. In addition, the proportion of valid data increases from 58.3% to 89.6%, demonstrating a significant improvement in data reliability and the overall quality of training trajectories.

## 3.3 PERFORMANCE ON DIFFERENT BENCHMARKS

### 3.3.1 INTERNAL OFFLINE WEBSITE BENCHMARK

We further evaluate models on an internal offline website benchmark that covers both operational tasks and information retrieval. As summarized in Table 3, general-purpose vision–language models such as QwenVL2.5-3B and GPT-4o exhibit limited capability, with task completion rates (TCR) below 30%. In contrast, models trained with reinforcement learning demonstrate substantially stronger performance. GUI-R1-3B achieves high accuracy across both task types, and our WebFactory-3B model attains comparable results, with slightly higher efficiency and accuracy (e.g., 71.8% vs. 68.2% TCR and 87.6% vs. 85.3% accuracy on operational tasks). These findings highlight that training solely on synthetic data enables WebFactory-3B to reach performance levels on par with models trained with large-scale human annotations.

### 3.3.2 OFFLINE-TO-ONLINE TRANSFER

To assess generalization to real-world scenarios, we evaluate models trained offline on three online platforms: Amazon, Airbnb, and Booking. As reported in Table 4, general-purpose models such as QwenVL2.5-3B and GPT-4o show limited transfer capability, with average task completion rates (TCR) below 40%. In contrast, reinforcement learning–based agents achieve markedly better perfor-

Table 4: Performance on offline-to-online transfer across Amazon, Airbnb, and Booking, reported by task completion rate (TCR) and accuracy.

| Model | Amazon | | Airbnb | | Booking | | Avg. |
|---|---|---|---|---|---|---|---|
| | TCR(%) | Acc.(%) | TCR(%) | Acc.(%) | TCR(%) | Acc.(%) | TCR(%) |
| QwenVL2.5-3B | 22.3 | 48.6 | 18.7 | 43.2 | 20.1 | 45.8 | 20.4 |
| GPT-4o | 41.2 | 68.7 | 37.8 | 64.3 | 39.6 | 66.2 | 39.5 |
| GUI-R1-3B | 38.6 | 65.3 | 35.2 | 61.7 | 37.1 | 63.4 | 37.0 |
| WebFactory-3B | **55.7** | **79.3** | **51.2** | **75.6** | **53.3** | **77.4** | **53.4** |

Table 5: Generalization on public GUI benchmarks (GUI-Act-Web and GUI-Odyssey), reported by type accuracy, grounding recall (GR), and success rate (SR). Best results are in bold.

| Setting | Model | GUI-Act-Web | | | GUI-Odyssey | | |
|---|---|---|---|---|---|---|---|
| | | Type | GR | SR | Type | GR | SR |
| Zero-Shot | GPT-4o | 77.1 | 45.0 | 41.8 | 37.5 | 14.2 | 5.4 |
| | QwenVL2.5-3B | 54.9 | 63.5 | 55.6 | 38.4 | 27.2 | 27.2 |
| RL Fine-Tuning | GUI-R1-3B | **89.9** | **87.4** | 76.3 | 54.8 | 41.5 | **41.3** |
| | WebFactory-3B | 89.0 | 82.1 | **84.2** | **66.0** | **48.1** | 40.9 |

mance. WebFactory-3B attains an average TCR of 53.4%, representing a 162% improvement over QwenVL2.5-3B (20.4%) and a 44% gain over GUI-R1-3B (37.0%). Furthermore, WebFactory-3B consistently achieves the highest accuracy across all three platforms (79.3% on Amazon, 75.6% on Airbnb, and 77.4% on Booking), underscoring its ability to transfer effectively from synthetic offline training to previously unseen online environments.

### 3.3.3 PUBLIC GUI AGENT BENCHMARKS

Performance on public benchmarks further validates our approach's effectiveness. As shown in Table 5, WebFactory-3B achieves strong generalization across diverse GUI benchmarks. On **GUI-Act-Web**, it obtains the highest success rate (SR) of 84.2%, surpassing both GPT-4o (41.8%) and QwenVL2.5-3B (55.6%). Although GUI-R1-3B yields slightly higher grounding accuracy (GR) on this benchmark (87.4% vs. 82.1%), WebFactory-3B consistently delivers better overall task completion.

On **OmniAct-Desktop**, WebFactory-3B attains a balanced performance with 85.3% Type accuracy and 73.9% SR, closely matching GUI-R1-3B while significantly outperforming zero-shot foundation models. Most notably, on the challenging **GUI-Odyssey** benchmark, WebFactory-3B reaches 66.0% Type accuracy, substantially higher than GUI-R1-3B (54.8%), GPT-4o (37.5%), and QwenVL2.5-3B (38.4%). This highlights its robust cross-domain transfer capability, even though it was trained solely on synthetic data. Overall, these results confirm that WebFactory-3B not only generalizes well but also provides consistent improvements across heterogeneous GUI environments.

### 3.4 PIPELINE PERFORMANCE WITH DIFFERENT FOUNDATION MODELS

To examine the generalizability of our pipeline and evaluate the *LLM embodiment* of different foundation models, we employ three state-of-the-art LLMs—GPT-5, Claude Opus 4.1, and Claude Sonnet 4—to drive the entire data generation process. Each model functions as the architect throughout the entire pipeline: from synthesizing the offline website environments via code generation, to formulating tasks, and finally collecting interaction trajectories. The resulting agents are subsequently evaluated on a diverse suite of benchmarks.

As shown in Figure 3, GPT-5 achieves the strongest overall performance, particularly excelling in Type accuracy while maintaining robust performance across diverse GUI environments. Claude Opus 4.1 performs competitively, yielding slightly lower yet stable results. In contrast, Claude Sonnet 4 demonstrates greater variability across benchmarks, indicating less consistent generalization ability.

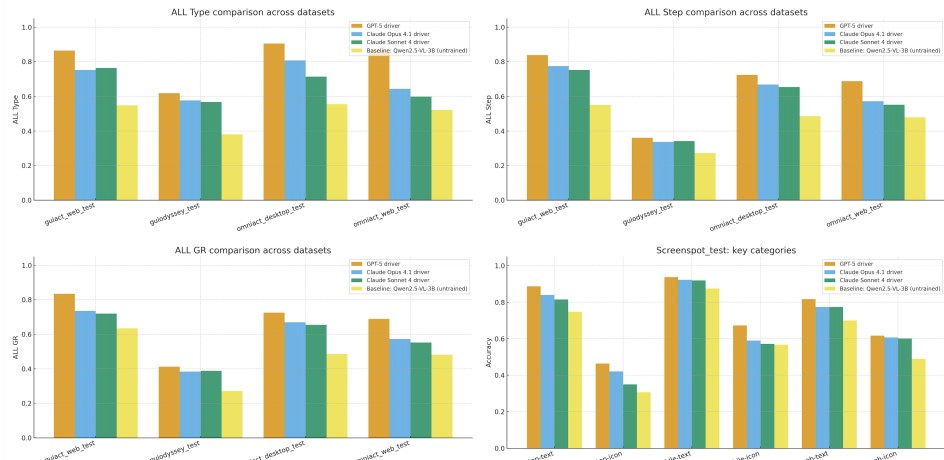

Figure 3: Performance comparison of agents trained with data generated by different foundation models across public GUI benchmarks. Results show Type accuracy, Step completion rate, and Grounding accuracy across GUI-Act-Web (Chen et al., 2024), GUI-Odyssey (Lu et al., 2024), OmniAct-Desktop (Kapoor et al., 2024), OmniAct-Web tests, and ScreenSpot categories (desktop-text, desktop-icon, mobile-text, mobile-icon, web-text, web-icon). GPT-5 consistently achieves the highest performance across most metrics, demonstrating superior data generation quality and intelligence compression capability.

## 4    DISCUSSION

Our agent's superior performance is more than an engineering success; it provides compelling evidence for our central thesis of intelligence compression. The proposed factory pipeline effectively demonstrates how to distill the vast, descriptive knowledge of LLMs into robust, actionable policies, outperforming even agents trained on extensive human data. This success underscores the decisive role of the LLM foundation model itself. Our findings reveal that a model's inherent reasoning and world knowledge directly cap the potential of the final agent, suggesting that the "transferability" and "embodiment potential" are critical, yet underexplored, dimensions for evaluating and selecting foundation models.

These insights motivate a necessary refinement of scaling laws for embodied agents. Analogous to LLM scaling laws, an agent's asymptotic performance may be governed not by raw data volume, but by a foundation model's intelligence compression efficiency and its inherent capability for embodiment. Our pipeline represents a first step in this direction, paving a path toward agents that can rapidly adapt and self-evolve in novel GUI environments by generating their own curricula. While validated here in GUI settings, we believe this paradigm of transforming latent knowledge into grounded action holds strong promise for more complex physical embodied environments.

**Future Work.**    Building on the pipeline's programmability, a promising avenue for future work is to leverage WebFactory for **targeted capability evolution**. Unlike static datasets, our generative infrastructure allows for the systematic probing of specific agent weaknesses—such as precise continuous interactions or complex logic handling—followed by the on-demand synthesis of dedicated website environments to address these deficits. This closed-loop mechanism, capable of identifying gaps and algorithmically generating the necessary embodied experiences to fill them, transforms the system into a self-correcting engine, further establishing the foundation for truly autonomous and robust agent intelligence.

**Limitations.**    While our pipeline demonstrates strong empirical results, we identify two primary avenues for future work. First, our work does not include an exhaustive ablation on the impact of different reward mechanisms. A deeper analysis comparing our decomposed reward against sparser or even LLM-generated reward functions could yield further insights into learning dynamics and final policy robustness. Second, *WebFactory* pipeline's performance in fundamentally different GUI

paradigms (e.g., game engines or specialized creative software) remains to be systematically validated. Exploring these directions will be crucial for assessing the true generality of our approach.

## 5 CONCLUSION

**WebFactory** demonstrates that high-fidelity offline environments, combined with knowledge-driven task generation and automated RL training, can produce web agents that transfer effectively to live websites. By eliminating the brittleness of online experimentation while preserving real-world complexity, our framework enables reproducible, scalable research. The open-source release of all components—websites, generators, training pipeline, and evaluation tools—provides a foundation for the community to build upon. As the intelligence of foundation LLMs increases and their costs decrease, we expect this offline-to-online, intelligence compression paradigm to become an increasingly practical path to capable, general-purpose web agents.

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

## A    DISCLOSURE OF LLM USE.

We used large language models to assist with language polishing and discovering related work. All technical claims, experiments, and analyses were designed, executed, and verified by the authors.

## B    RELATED WORK

**Web environments and benchmarks.**    Research on web-agent environments has gradually evolved from simplified DOM-centric tasks to realistic, multi-domain benchmarks. Early controlled settings such as MiniWoB and MiniWoB++ provided reproducible yet toy-scale interactions for evaluating RL policies (Shi et al., 2017; Liu et al., 2018). Subsequent efforts increased realism: VisualWebArena added multimodal grounding to web-page interactions (Koh et al., 2024), while WebArena introduced high-fidelity, self-hostable environments covering e-commerce, forums, software development, and content management with integrated tool support (Zhou et al., 2024). WebChoreArena further emphasized long-horizon reasoning and reproducibility by designing hundreds of durable, labor-intensive tasks (Miyai et al., 2025). Beyond browser-centric setups, simulation-based environments such as ALFWorld and BabyAI studied language-grounded, partially observed control problems, offering insights on curriculum learning and generalization (Shridhar et al., 2020; Chevalier-Boisvert et al., 2018). Unlike live-web evaluations, which are hindered by CAPTCHAs, layout drift, and network nondeterminism, *WebFactory* adopts a *versioned, fully offline, pre-authenticated* design with deterministic rendering and explicit knowledge/data snapshots, enabling *verifiable answers* and *replayable trajectories* at scale.

**Frameworks and datasets.**    A parallel line of work has focused on standardizing interfaces and expanding coverage. BrowserGym unifies APIs across multiple environments (MiniWoB, VisualWebArena, WebArena), facilitating consistent comparison of agents (Chezelles et al., 2024). Mind2Web aggregates thousands of human-annotated tasks across diverse websites, emphasizing breadth and realistic natural-language instructions (Deng et al., 2023). Other benchmarks extend to OS- and GUI-level control, such as Windows Agent Arena, OSWorld, Android-in-the-wild, and GUI-Odyssey, which target distinct substrates and I/O stacks beyond the browser (Bonatti et al., 2024; Abhyankar et al., 2025; Rawles et al., 2023; Lu et al., 2024). In contrast, *WebFactory* focuses on *browser-native* interaction under complete controllability, while remaining compatible with common evaluation interfaces for cross-benchmark comparison.

**Training paradigms for web agents.**    Recent work has explored both supervised sequence modeling and RL-based methods tailored to long-horizon web interaction. Decision Transformer applies return-conditioned sequence modeling to agent trajectories (Chen et al., 2021); conservative offline RL enhances safety when learning from static datasets (Kostrikov et al., 2021); and preference- or feedback-driven optimization aligns policies with human intent (Christiano et al., 2017; Ouyang et al., 2022). Web-specific innovations include curricula derived from agent failure modes and outcome-supervised reward modeling (Qi et al., 2024), success-driven rollouts (Wei et al., 2025), reusable skill abstractions (Zheng et al., 2025), and hierarchical formulations for decomposing complex browsing workflows into subgoals (Furuta et al., 2023). *WebFactory* complements these paradigms by offering a scalable pipeline where synthetic trajectories, unified action spaces, and decomposed rewards can be directly applied to train robust web agents.

**Reasoning, exploration, and data collection.**    Reasoning scaffolds such as ReAct, Voyager, Reflexion, and Tree-of-Thoughts enhance planning, self-correction, and exploration (Yao et al., 2023b; Wang et al., 2023; Shinn et al., 2023; Yao et al., 2023a). On the data side, Go-Browse uses graph-guided exploration to diversify trajectories (Gandhi & Neubig, 2025); WebVoyager retrospectively synthesizes demonstrations from failures without human annotations (He et al., 2024); and AgentOccam streamlines observation–action design to align with LLM reasoning (Yang et al., 2024). Curriculum-based difficulty further adapts training to agent errors (Qi et al., 2024). *WebFactory* provides a reproducible substrate—explicit tasks, normalized answers, and deterministic replays—to evaluate reasoning/exploration methods and generate large, high-signal offline datasets without costly online rollouts.

**Live-web automatic task/trajectory generation.** Synatra converts human-oriented tutorials and indirect instructions into synthetic, executable demonstrations for web agents, enabling large-scale supervision without manual trajectory annotation (Ou et al., 2024). Harnessing Webpage UIs builds a large text-rich visual understanding dataset from real webpages, exploiting UI structure as a multimodal supervision signal (Liu et al., 2024). PAE (Proposer-Agent-Evaluator) introduces a proposer–agent–evaluator loop that autonomously proposes web tasks, executes them, and filters trajectories to discover reusable skills for foundation-model agents (Zhou et al., 2025). NNetNav leverages unsupervised interaction with live websites together with hindsight relabelling to construct browser-agent training data directly from in-the-wild behavior (Murty et al., 2024). InSTA pushes this line to internet scale, generating and judging tasks and trajectories across a large number of live websites with LLM-based agents and evaluators (Trabucco et al., 2025). These systems collectively excel in scale and real-world diversity on the live web. In contrast, *WebFactory* operates on a fully offline, versioned web suite with deterministic rendering, explicit knowledge/data snapshots, verifiable optimal paths, and replayable trajectories, prioritizing determinism, precise reward specification, and reproducibility, and thus providing a complementary substrate to live-web pipelines rather than a replacement.

**Weak/indirect-knowledge–to-trajectory pipelines.** WebShop formulates a goal-conditioned shopping environment on real e-commerce sites, using product metadata and textual descriptions to supervise grounded navigation and decision making (Yao et al., 2022). Synatra converts human-oriented web tutorials and other indirect instructions into large-scale executable demonstrations, providing high-coverage synthetic supervision for web agents (Ou et al., 2024). AgentTrek synthesizes trajectories by guiding replay with web tutorials as high-level instruction sequences, tightening the link between weak textual knowledge and concrete action sequences (Xu et al., 2024). Explorer scales exploration-driven web trajectory synthesis across many real webpages, using multimodal exploration signals to expand demonstration coverage for multimodal web agents (Pahuja et al., 2025). These pipelines demonstrate how weak or indirect web knowledge can be systematically transformed into training data on the live web. *WebFactory* instead operates in a fully controllable offline suite with deterministic rendering, structured knowledge snapshots, and verifiable optimal paths, offering a reproducible substrate that is complementary to these weak-supervision pipelines.

## C  BUILDING THE OFFLINE WEBSITE SUITE

### C.1  DESIGN GOALS

We target a high-fidelity yet fully controllable suite that (i) boots to pre-authenticated, seeded sessions, (ii) exposes ground-truth knowledge and data for generation and evaluation, (iii) disables anti-automation friction (CAPTCHA, bot detection), and (iv) is versioned and reproducible.

### C.2  LLM-DRIVEN BUILD PROCESS

We have developed a method for scalable, high-fidelity offline website generation using LLMs, which we plan to open source to facilitate reproducibility and community extension. The construction of each offline website is fully automated: the "site recipe" is executed by LLM-driven coding agents. WebFactory acts as an extensible engine where LLMs function as embodied architects, enabling scalable generation of high-fidelity web environments.

Our automated build process follows a uniform site recipe across domains (e-commerce, travel, *etc.*):

1. **Scaffold & theming.** Initialize a Next.js/React monorepo with a shared UI kit (forms, tables, modal, hover menus, drag-and-drop). Provide mobile/desktop breakpoints to produce realistic layout variety.
2. **Data layer materialization.** For each site family, export a versioned static snapshot (`Data.js`/JSON) with deterministic seeds. Schema includes entities (e.g., `Product`, `Hotel`, `Flight`, `Message`), relations, and canonical views (list/detail/cart/checkout).
3. **Navigation graph & flows.** Encode page graph and canonical flows (e.g., browse → detail → cart) in a machine-readable `knowledge.json`. Each node stores visible affordances and element locators.

4. **Anti-bot off-switch.** Gate any bot-detection middleware by a build flag; fall back to no-challenge in offline mode.

5. **Benchmark export.** For each version $v$, release (i) site bundle, (ii) knowledge.json, (iii) Data.json, and (iv) scripted evaluators.

Table 6: Overview of offline website families used in our benchmark, including their domains and representative core functionalities.

| Name | Domain | Core Functionality |
|---|---|---|
| Shopping | E-commerce marketplace | Marketplace with search, multi-facet filters, cart/wishlist, reviews, multi-step checkout. |
| Mealdash | Food delivery | Restaurant discovery, dietary filters, cart with quantity/notes, scheduling, order tracking. |
| Hotels | Hotel booking | Location/date/guest search, amenity filters, room types, price tiers, availability & reservation. |
| Flights | Flight search & booking | One-way/return/multi-city, flexible dates, carriers/cabin filters, fare comparison, seat selection. |
| Careerlink | Professional networking & jobs | Job search by skills/company, profiles, applications, resume management, insights. |
| Carrental | Car rental | Pickup/dropoff, vehicle class, insurance add-ons, driver requirements, booking changes. |
| Masterticket | Event ticketing | Artist/venue search, event categories, date filters, seat map, ticket types, fees, checkout. |
| Staybnb | Short-term rentals | Rentals by location/dates/guests, amenity filters, calendars, pricing, booking flows. |
| Email | Email client | Folders, compose/reply/forward, attachments, rules, search, threads, contacts. |
| Companycheck | Company data & intelligence | Company profiles, filings, executives, relationship graph, compliance & due diligence. |

# D   TASK SYNTHESIS DETAILS

## D.1   DUAL-TRACK GENERATION

**(1) Template-driven** Define modular spaces for search, filtering, sorting, cart/checkout, form completion, multi-form workflows, and cross-page navigation. Instantiate by sampling from versioned Data.* with constraints (*e.g.*, "price $\leq$ \$200", "rating $\geq$ 4") while respecting site schemas.

**(2) LLM-assisted** Feed compact knowledge slices—navigation graph, page affordances, canonical flows—and sampled data skims to an LLM to propose task *paths* beyond the template envelope (long-horizon operations and compositional IR).

## D.2   VALIDATORS & EXECUTABILITY

All candidates pass:

- **Schema conformance**: fields and operators exist in Data.json.
- **Visibility check**: referenced elements/records are reachable and visible given viewport and filters (uses layout probes).
- **Path feasibility**: dry-run along the known navigation path (knowledge.json); failure short-circuits.
- **Answerability (IR)**: canonical answers exist in the data layer with a unique normalized target.

## D.3   DIFFICULTY CONTROL & CURRICULUM

We sample along three axes: data complexity (catalog/graph density), UI complexity (multi-level nav, drag-and-drop, hover), and workflow depth (lookup $\rightarrow$ multi-step execution). Curricula ramp

---

**Algorithm 1** Knowledge-driven Task Factory

---

1: **for** site $s$ in sites **do**
2:     $K \leftarrow$ load_knowledge($s$);   $D \leftarrow$ load_data($s$)
3:     **for** spec in templates $\cup$ llm_prompts **do**
4:         $cands \leftarrow$ instantiate(spec, $D$, difficulty)
5:         **for** $t$ in $cands$ **do**
6:             **if** schema_ok($t$, $D$) && visible($t$, $K$) && path_feasible($t$, $K$) **then**
7:                 attach_gold_path($t$, $K$); **emit** $t$
8:             **end if**
9:         **end for**
10:     **end for**
11: **end for**

---

(i) start URLs, (ii) number of required filters, (iii) cross-page hops, (iv) time/step budget. Each emitted task is stored with a difficulty tag and *gold* path.

# E    TRAJECTORY GENERATION DETAILS

## E.1    EXECUTOR & INSTRUMENTATION

We execute the pre-validated task set $\mathcal{T}$ with a strong executor (OpenAI `computer-use-preview`) inside the offline suite. Each step logs:

- page ID, viewport hash, DOM key-node set;
- action tuple $\mathbf{a}_t = \{a_t^{act}, a_t^{point}, a_t^{text}\}$ and matched locator;
- state diff summary (element attributes, cart contents, form values).

Screenshots and structured traces are stored in Parquet; per-episode metadata carries seed, site version, and curriculum tier.

## E.2    FILTERING & DETERMINISM

We remove low-quality traces via:

1. **Deterministic replay**: re-run with the same seed; reject if hashes (viewport, key-node set, cart snapshot) mismatch.
2. **Key-node coverage**: ensure required nodes along the gold path are visited in the right order.
3. **IR validation**: compute normalized $F_1$ against canonical answer; drop if below threshold.

Accepted trajectories populate the replay buffer $\mathcal{B}$ with tuples $(s_t, \mathbf{a}_t, R_t, s_{t+1})$.

# F    TRAJECTORY DATASET STATISTICS & DISTRIBUTIONS

To better understand the dynamics of user interactions within the trajectory dataset, we analyze both the overall action distribution and the transition patterns between different actions.

## F.1    ACTION DISTRIBUTION

Figure 4 presents the distribution of ground-truth actions. The dataset is dominated by `click` actions, which account for nearly half of all recorded interactions (47.8%). This is followed by `wait` (24.1%) and `scroll` (20.9%), reflecting common patterns in typical graphical user interface (GUI) usage. Less frequent actions include `type` (5.3%), `keypress` (1.8%), and `double_click` (0.2%). These results suggest that the dataset is heavily skewed towards basic navigational primitives (click, wait, and scroll), which together comprise over 90% of the interactions.

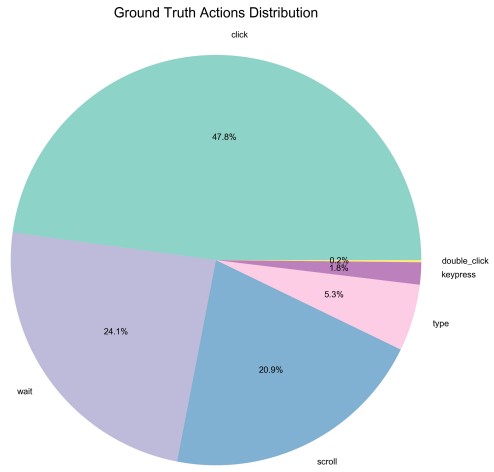

Figure 4: Ground truth action distribution in the dataset.

## F.2 ACTION TRANSITION DYNAMICS

To capture sequential dependencies, we compute the transition frequency between all pairs of actions (Figure 5). The heatmap reveals several key patterns:

- `click` frequently transitions back to itself (812 times), and is also followed by `wait` (551) and `scroll` (191). This indicates that clicking is often interleaved with periods of waiting or subsequent navigation.
- `scroll` transitions strongly to itself (419) and also to `click` (289), reflecting the natural alternation between scrolling content and selecting items.
- `wait` is another central action, often followed by `click` (480) and self-repetition (241), which captures idle or delay states before further activity.
- Rare transitions occur for `double_click` and `keypress`, consistent with their overall low frequency.

Together, these findings highlight that the dataset is structured around a small number of dominant action primitives, with strong self-loops and predictable sequential dynamics. Such patterns are valuable for modeling purposes, as they suggest that predictive models may benefit from emphasizing high-frequency transitions while carefully handling the long-tail actions.

## G   RL TRAINING DETAILS

### G.1   ACTION SPACE (WEB-SPECIFIC)

We operate in a structured space:

$$\mathbf{a}_t = \{a_t^{\text{act}}, a_t^{\text{point}}, a_t^{\text{text}}\},$$

$$a_t^{\text{act}} \in \{\text{click}, \text{double\_click}, \text{type}, \text{scroll}, \text{keypress}, \text{drag}, \text{get\_final\_answer}\}.$$

A concise specification is provided in Table 7.

### G.2   REWARD

Per-step reward:

$$R_t = \alpha R_f + \beta R_{\text{accuracy}}, \tag{4}$$

where $R_f$ enforces structured outputs (valid JSON, tags, type constraints) and $R_{\text{accuracy}}$ is action-specific:

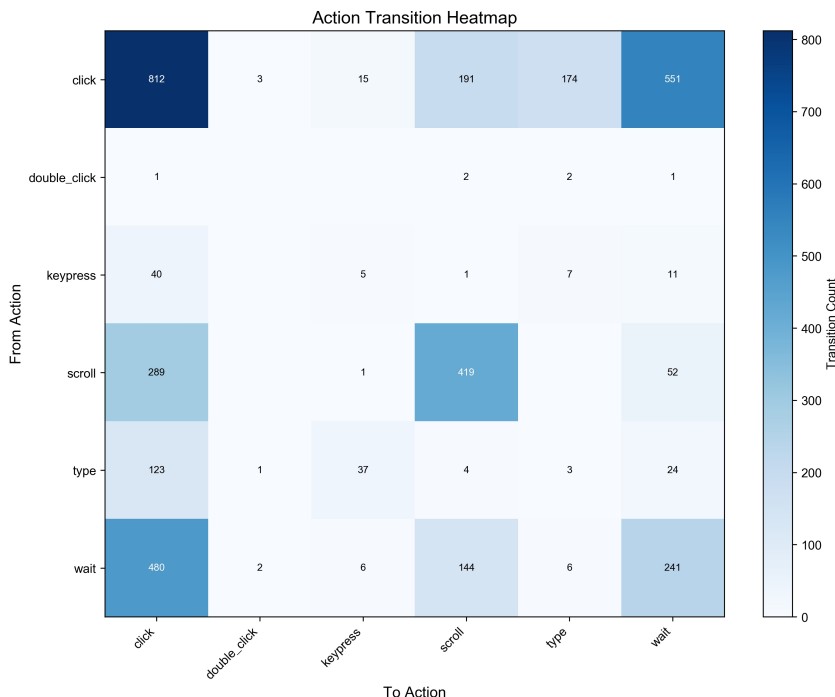

Figure 5: Action transition heatmap showing transition counts between actions.

Table 7: Detailed specification of the web agent action space, listing each supported action together with its required point parameters (coordinates) and text parameters.

| Action | Point Parameter | Text Parameter |
|---|---|---|
| click | $[x, y]$ | – |
| double_click | $[x, y]$ | – |
| type | $[-100, -100]$ | input text (required) |
| scroll | $[-100, -100]$ | UP/DOWN |
| keypress | $[-100, -100]$ | key name (e.g., ENTER) |
| drag | $[[x_1, y_1], [x_2, y_2]]$ | UP/DOWN |
| get_final_answer | $[-100, -100]$ | final answer text |

- **Click family** (click/double_click): inside-target check + distance tolerance (140px) to element center.

- **Text family** (type/keypress): token-level $F_1$ with case/punctuation normalization; single-token special-cased.

- **Scroll/drag**: exact direction match; drag validates source–target coordinates.

- **IR answer**: final answer scored by normalized $F_1$ against canonical target.

### G.3 TRAINING ALGORITHM (GRPO/PPO)

We extend GUI-R1 with an IR-aware head and reward. GRPO is used for group-normalized advantages over multi-sample rollouts.

---

**Algorithm 2** GRPO for Web Agents

---

 1: **for** episode $e = 1$ to $E$ **do**
 2:   Sample task batch $\{p_i\}$
 3:   **for** each $p_i$ **do**
 4:     Generate $n$ trajectories; compute $R_{i,1..n}$
 5:     $\mu_i \leftarrow \text{mean}(R_{i,*}),\ \sigma_i \leftarrow \text{std}(R_{i,*})$
 6:     $A_{i,j} \leftarrow (R_{i,j} - \mu_i)/(\sigma_i + \epsilon)$
 7:   **end for**
 8:   Update $\pi_\theta$ with PPO loss and KL penalty
 9:   **if** $e \bmod s = 0$ **then**
10:     save checkpoint
11:   **end if**
12: **end for**

---

## H  IMPLEMENTATION DETAILS

### H.1  TRAINING INFRASTRUCTURE

**Model.** Qwen2.5-VL-3B; vision encoder initially frozen; max screenshot $1258 \times 1258$.
**Distributed training.**

- Actors: 4 GPUs with FSDP; rollouts: 1 GPU via vLLM.
- Global batch 64, micro-batch 4; optimizer states CPU offloaded.

**Optimization.** AdamW, lr $1 \times 10^{-6}$, wd 0.01, grad clip 1.0, fixed KL 0.01.

### H.2  WEB AGENT DATA PROCESSING

**Parquet traces** contain (i) screenshots + action history, (ii) task instruction, (iii) ground-truth actions with bboxes, (iv) task-type flags.

### H.3  REWARD COMPUTATION FOR WEB TASKS

See Sec. G for per-action rules; format reward enforces valid JSON, required <think>/<answer> tags, numeric types, and conditional fields.

Table 8: Hyperparameters used for web agent RL training.

| Parameter | Value |
|---|---|
| Episodes $E$ | 15 |
| Rollout temperature | 1.0 |
| Responses per prompt $n$ | 5 |
| Global / micro batch | 64 / 4 |
| Max prompt / response tokens | 2048 / 1024 |
| Image resolution | $1258 \times 1258$ |
| PPO clip | 0.2 |
| Discount $\gamma$ | 1.0 |
| $\alpha$ (format) / $\beta$ (accuracy) | 0.2 / 0.8 |
| KL penalty | 0.01 |
| Learning rate | $1 \times 10^{-6}$ |
| GPUs | $4 \times$ V100/A100 |

# I   OFFLINE-TO-ONLINE TRANSFER EVALUATION DETAILS

In this section, we provide a comprehensive specification of the experimental protocol used for the Offline-to-Online Transfer evaluation (Table 4 in the main text). To ensure rigorous validation of agent capabilities in live, dynamic environments, we constructed a custom benchmark of 90 live tasks across Amazon, Airbnb, and Booking.

## I.1   METHODOLOGY FOR UNBIASED TASK CONSTRUCTION

To prevent selection bias and ensure the benchmark accurately reflects real-world usage, we adhered to a "User-First, Model-Agnostic" design protocol. Tasks were defined based on necessary user workflows prior to any agent evaluation.

- **Funnel-Based Coverage:** We strictly stratified tasks by user conversion stages to ensure full spectrum coverage:
  - *Discovery:* Ambiguous search queries requiring exploration.
  - *Refinement:* Complex filtering involving price ranges, brands, and amenity constraints.
  - *Action:* State-changing operations such as adding items to a cart or selecting specific dates.
- **Complexity Alignment:** We deliberately excluded trivial single-step tasks. All tasks align with "Medium" (3–5 steps) and "High" (>5 steps) complexity tiers to force long-horizon reasoning.
- **Cross-Domain Mapping:** Online platforms were selected to strictly correspond to our offline training domains to evaluate transfer capability: Amazon $\rightarrow$ Shopping, Airbnb $\rightarrow$ StayBnB, and Booking $\rightarrow$ Hotels.

## I.2   EXECUTABILITY AND STABILITY VERIFICATION

Live websites are subject to frequent changes, A/B testing, and inventory fluctuations. To rule out failures caused by external factors (e.g., out-of-stock items or UI updates), all tasks are manually verified for feasibility within 24 hours prior to agent evaluation. This ensures that reported failures are due to agent capability rather than environmental errors.

## I.3   STATISTICAL SIGNIFICANCE ANALYSIS

Given the sample size of $N = 90$ (30 tasks per platform), we calculated the 95% Confidence Intervals (CI) for the success rates using the Wilson Score Interval method, which is robust for smaller sample sizes.

- **Baseline (QwenVL2.5-3B):** 20.4% [95% CI: 13.0% – 30.0%]
- **WebFactory-3B (Ours): 53.4%** [95% CI: 43.0% – 63.5%]

Notably, there is **no overlap** between the confidence intervals. The lower bound of our method (43.0%) is substantially higher than the upper bound of the baseline (30.0%). This statistical analysis confirms that the 162% relative improvement reported in the main text is robust and significant, rather than a result of sampling noise.

## I.4   REPRESENTATIVE TASK SPECIFICATIONS

Table 9 provides representative examples of the tasks used in the evaluation, detailing the required interactions and complexity constraints.

**Conclusion on Rigor.**   The tasks detailed above involve Modal Interactions, Calendar Pickers, and Multi-step Filtering—complex UI patterns that standard multimodal models (e.g., QwenVL2.5, 20.4% SR) struggle to handle zero-shot. The 53.4% success rate of WebFactory-3B, supported by

Table 9: Representative tasks from the Offline-to-Online Transfer Benchmark. Tasks are designed to test specific interaction capabilities including precise search, constraint filtering, and complex state changes.

| Platform | Category | Design & Complexity Constraints | Representative Instruction |
|---|---|---|---|
| **Amazon** | A: Precise Search | **Req:** Precise keyword matching; navigate to detail page to verify specs. **UI:** Search bar, Click. | "Search for 'Sony WH-1000XM5 headphones', click on the black version, and tell me the current price." |
| | B: Constraint Filtering | **Req:** Apply multiple compound filters (price, brand, Prime) before clicking result. **UI:** Sidebar filters. | "Find a 'Gaming Monitor' under $300 from 'ASUS' with '144Hz' refresh rate. Add the first result to the cart." |
| | C: Cart Flow | **Req:** State changes; multi-page navigation. **UI:** Add-to-cart, verify cart. | "Add a 'Logitech MX Master 3S' to your cart, then go to the cart and change the quantity to 2." |
| **Airbnb** | A: Date/Loc Search | **Req:** Complex calendar interaction; location input. **UI:** Date-picker, search field. | "Search for a stay in 'Kyoto, Japan' for '2 adults' from 'September 10' to 'September 15'." |
| | B: Amenity Filtering | **Req:** Open modal window; scroll to find/check specific boxes. **UI:** Modal popups, checkboxes. | "Find a home in 'Paris' that has 'Wifi', 'Kitchen', and 'Washing Machine'. Open the detail page of the highest-rated listing." |
| | C: Detail Extraction | **Req:** Parse unstructured long-text descriptions. **UI:** Text parsing, scrolling. | "Go to the first listing for 'Cabin in Lake Tahoe' and verify if 'Pets are allowed' in the house rules." |
| **Booking** | A: Multi-Criteria | **Req:** Handle location, dates, room/guest config simultaneously. **UI:** Complex form filling. | "Search for a hotel in 'New York' for '2 adults, 1 child' for the weekend of 'September 14th'." |
| | B: Sort & Select | **Req:** Operate sorting dropdowns; select under constraints. **UI:** Dropdowns, list parsing. | "Sort hotels in 'London' by 'Top Reviewed' and click on the hotel with the highest score under £200/night." |
| | C: Room Config | **Req:** Navigate to detail page; extensive scroll; specific selection. **UI:** Deep navigation. | "Search for 'Hilton Tokyo', scroll to available rooms, and select a 'King Room' with 'Breakfast Included'." |

non-overlapping confidence intervals, provides strong evidence that our "Intelligence Compression" paradigm successfully transfers generalizable logic from controlled offline data to unseen, noisy online environments.

