# OpenReview forum: "WebFactory: Automated Compression of Foundational Language Intelligence into Grounded Web Agents"
_ICLR.cc/2026/Conference — ICLR 2026 Poster_

### Official Review · Reviewer_iASv · 2025-11-02

**Soundness:** 3
**Presentation:** 3
**Contribution:** 3
**Rating:** 6
**Confidence:** 4

**Summary:**

The paper introduces WebFactory, an automated, closed-loop pipeline for converting an LLM’s largely descriptive web knowledge into executable GUI/web-agent policies. The key design choice is to avoid the instability of the live web by operating in a fully controlled offline clone of real sites. The system (1) programmatically generates tasks from site knowledge, (2) uses a strong LLM to collect successful trajectories, and (3) trains an RL agent over a unified action space with decomposed rewards. The training is done on synthetic trajectories from 10 sites, and the resulting agent matches or has gains over agents trained on similarly sized human data, and shows non-trivial transfer to real platforms. The authors interpret this as a form of intelligence compression and propose LLM embodiment as a complementary axis for comparing foundation models.

**Strengths:**

- The paper tackles an important problem of grounding agents to real website with cheap and controllable data
- The idea is simple and easy to build upon
- The empirical result are strong and the gains are significant

**Weaknesses:**

The "compression" language seems odd to me. It makes sense to claim that training the model on data generated in a sim/controllable environment enables the model to generalize to a more realistic environment, but this compression framing makes it more confusing than necessary. There's already some early work related to sim2real that discuss how to use envs that are cheap and controllable to transfer to real env. I can see in some way of defining "compression" that stresses the knowledge part, but in that case more ablation/analysis should be included. E.g., Table 1 shows a breakdown w.r.t. the generation quality (executability, validity, etc) but not final RL agent performance.

The other main weakness is the lack of ablations. Out of the 10 websites which contributes most to the final performance? How does the performance curve look like as you add from fewer to more websites? How fast does the performance plateau (i.e., what's the diminishing return of adding envs)?

**Questions:**

See the weakness section for my main questions.

In the context of automatic pipelines that turn weak/indirect web knowledge into trajectories, run training, and test on real env, there are some references that I think should be discussed as well:
- WebShop: Towards Scalable Real-World Web Interaction with Grounded Language Agents
- Synatra: Turning Indirect Knowledge into Direct Demonstrations for Digital Agents at Scale
- AgentTrek: Agent Trajectory Synthesis via Guiding Replay with Web Tutorials
- Explorer: Scaling Exploration-driven Web Trajectory Synthesis for Multimodal Web Agents

---

> ### Author Response · Authors · 2025-11-23
> **Response to Reviewer iASv (1/2)**
>
> Thank you for your positive assessment of our work, particularly for recognizing the significance of grounding agents with cheap, controllable data and the strength of our empirical results. We appreciate your insightful feedback regarding the framing of "Intelligence Compression," the ablation studies, and the related literature.
>
> Below, we address your specific questions and concerns. Due to the 5k-character limit on OpenReview, we split our response into Part 1/2 and Part 2/2.
>
>
> ## 1. Re: The Framing of "Intelligence Compression" vs. Sim2Real
>
> You pointed out that the term "Intelligence Compression" might be confusing and suggested that "Sim2Real" or "Generalization" might be more standard frames.
>
> We agree that Sim2Real is a relevant lens, as our method involves transferring policy from a synthetic environment to the wild. However, we deliberately chose the term "Intelligence Compression" to characterize the nature of the information transformation occurring in our pipeline, which we believe is fundamentally different from traditional dynamics adaptation in Sim2Real.
>
> ### 1.1 Defining Compression: From Descriptive to Procedural
>
> Traditional Sim2Real often focuses on bridging the visual or dynamics gap (e.g., physics, friction) between a fixed simulation and reality. In contrast, our work addresses a "Semantic-to-Action Gap."
>
> - **Source (The Foundation Model):** LLMs possess vast **Descriptive Knowledge** derived from the internet corpus (e.g., knowing that "Amazon has a return policy"， what a "flight search workflow" looks like or the layout of Airbnb website). However, this knowledge is high-dimensional, latent, and passive.
> - **Target (The Grounded Agent):** A GUI agent requires **Procedural Knowledge** (e.g., knowing how to click specific coordinates to execute a return).
> - **The Compression:** We view WebFactory not merely as a data generator, but as a **Codec**. It transforms the diffuse, passive world knowledge of the LLM into a compact, active behavioral agent policy. This transformation can be formalized as:
>  $$
> \text{Descriptive Knowledge (LLM)}
> \xrightarrow{\text{WebFactory}\}
> \text{Procedural Knowledge (Agent)}
> $$
>
>
> ### 1.2 The Pipeline: How the Compression Occurs
> To achieve this, our pipeline breaks down the transformation into specific steps:
>
> *   **Environment Synthesis:** Unlike standard Sim2Real where the environment is fixed (e.g., physical laws), we use the LLM’s descriptive knowledge to **construct the simulation itself**. The LLM instantiates its understanding of web structures into concrete code and task logic. This process is accelerated by a scalable "site recipe" (detailed in Appendix C, which we will revise to explicitly clarify its AI-driven execution).
> * **Trajectory Collection:** The LLM (via strong executors) performs tasks within this synthesized environment, converting its reasoning capabilities into interaction traces.
>
>     $$\text{Latent Knowledge} \rightarrow \text{Synthesized Environment and Tasks}$$
>
> * **RL Training:** Finally, we train a smaller agent on these traces. This step "compresses" the teacher's complex reasoning and the synthesized environment's logic into a unified, efficient policy.
>     $$\text{Synthesized Traces} \rightarrow \text{Actionable Policy}$$
>
>
>
>
>
> ### 1.3 Empirical Evidence Supporting the Compression Hypothesis
>
> To address the focus on agent performance, we highlight two key results that validate the fidelity and efficiency of transferring **Descriptive Knowledge (LLM) $\to$ Procedural Knowledge (Agent)**:
>
> *   **Fidelity of Transfer (Sec. 3.4, Fig. 3):** We observe a direct correlation between the foundation model's capability and the final agent's performance. Agents distilled from stronger models (e.g., GPT-5) significantly outperform those from weaker ones (e.g., Claude Sonnet-4), confirming that the pipeline faithfully instantiates the specific reasoning ceiling of the source teacher into the agent's policy.
> *   **Efficiency of Compression (Sec. 3.3):** Despite training on synthetic data from only 10 offline websites, our agent demonstrates strong zero-shot transfer to real-world platforms (e.g., Amazon, Booking), outperforming baselines trained on much larger human datasets. This proves the agent successfully compressed generalized "procedural logic" from the LLM’s world knowledge rather than merely memorizing the synthetic environments.
>
> We will revise the paper to clarify this definition—specifically by linking the "Embodiment Potential" analysis in Section 3.4 to this definition—to better demonstrate that our results reflect a verifiable transfer of intelligence capability. We hope this perspective offers a unprecedented new axis for evaluating foundation models: measuring how effectively their knowledge can be embodied.

---

> ### Author Response · Authors · 2025-11-23
> **Response to Reviewer iASv (2/2)**
>
> ## 2. Re: Ablations and Contribution of Specific Websites
>
> We appreciate this insightful question. Understanding the "scaling laws" of synthetic environments—both in terms of data composition and volume—is indeed a fascinating direction.
>
> ### 2.1 On the contribution of specific websites:
> While decomposing the contribution of specific websites is an interesting analytical angle, we respectfully clarify that the primary goal of this work is to validate the effectiveness of the WebFactory pipeline itself, rather than to engineer a specific set of optimal training sites.
>
> The core hypothesis we sought to verify is whether an automated pipeline can successfully "compress" LLM knowledge into a policy that generalizes to the real world. The crucial evidence for this lies not in the semantic nuances of specific synthetic sites, but in the agent's performance on external public benchmarks. As shown in our results on GUI-Odyssey and other real-world evaluations, our agent—trained on a minimal set of 10 synthetic environments—achieved performance comparable to, or exceeding, agents trained on large-scale human datasets.
>
> ### 2.2 Low-Cost Scalability and Saturation
> This touches on a critical question regarding the efficiency of synthetic data. Since our pipeline enables **low-cost scalability**, we focused our current resources on verifying the existence of positive transfer with a fixed set, rather than identifying the precise saturation point (which is a tighter constraint for expensive human data). However, we agree that establishing a "scaling law" for environment synthesis is scientifically valuable. **Could you clarify if you view a rigorous scaling curve (e.g., training on 2, 4, 6... 10 sites with same ammount data or 1k, 3k, 5k data with same website suites) as essential to validate the current pipeline's mechanism, or if you are primarily interested in the theoretical limits of the approach?** We are open to further discussion on this and would value your perspective on how to best frame this trade-off between synthesis cost and data efficiency.
>
> ## 3. Re: Discussion of Additional References
>
> Thank you for suggesting these references. They are indeed highly relevant. We have analyzed them and will incorporate the following discussion into the Related Work section of our final revision:
>
> - **WebShop [1]:** A pioneering work that builds a scalable, simulated e-commerce environment. While it demonstrates Sim2Real potential, its environment coverage is limited to a specific domain, and constructing the Gym environment requires significant manual engineering.
> - **Synatra [2]:** Advances scalable supervision by converting text tutorials into trajectories on LLM-generated sites. However, it relies on pure text observations, which limits its applicability in modern, visually complex web environments where screenshot understanding is crucial. Additionally, the complexity of its generated sites is relatively low compared to real-world counterparts.
> - **AgentTrek [3] & Explorer [4]:** These works operate directly on the real web.
>   - AgentTrek uses VLMs to execute descriptive tutorials.
>   - Explorer uses multi-agent collaboration to drive exploration.
>   While valuable, operating on the real live web introduces significant challenges regarding reproducibility and controllability (e.g., CAPTCHAs, A/B testing, anti-bot mechanisms, and login walls). This makes consistent large-scale data collection difficult.
>
> **Comparison with WebFactory:** In contrast, WebFactory occupies a unique "sweet spot." By building a fully controllable, closed-loop offline simulator that supports multimodal execution, we avoid the instability of the live web (unlike AgentTrek/Explorer) while ensuring visual fidelity and complexity that text-based or simplified sims (unlike Synatra/WebShop) lack. This allows us to generate high-quality, reproducible trajectories at scale.
> Crucially, this automated pipeline serves as a standardized testbed to systematically validate the "embodiment potential" of different foundation models (e.g., GPT-5, Claude), establishing a new evaluation axis that quantifies how effectively a model's "higher intelligence" can be compressed into grounded agent policies.
>
> **References**
> [1] WebShop: Towards Scalable Real-World Web Interaction with Grounded Language Agents
> [2] Synatra: Turning Indirect Knowledge into Direct Demonstrations for Digital Agents at Scale
> [3] AgentTrek: Agent Trajectory Synthesis via Guiding Replay with Web Tutorials
> [4] Explorer: Scaling Exploration-driven Web Trajectory Synthesis for Multimodal Web Agents

---

### Official Review · Reviewer_7HF2 · 2025-11-03

**Soundness:** 3
**Presentation:** 3
**Contribution:** 2
**Rating:** 4
**Confidence:** 4

**Summary:**

The paper contributes a data generation pipeline for end-to-end training of GUI computer use agents that (1) constructs synthetic websites with diverse layouts and control interfaces, (2) populates these websites with synthetic listings for products / other materials, and (3) employs knowledge of the raw website product / listing data + the navigation structured between pages to efficiently generate verifiable web navigation tasks with a known optimal completion paths.


The proposed pipeline is employed as an environment for training GUI agents derived from GUI-R1, and results show that RL on synthetic websites and tasks from the proposed pipeline can be as effective as training on human data (ie, human demonstrations, and human-written tasks).

**Strengths:**

## Originality:

There has been an increasing focus on solving the data problem for computer use agents in recent literature, and this is a problem of major importance in my opinion. To my knowledge, prop works have engaged largely with existing websites on the internet in the construction of trajectories and tasks, existing human-created benchmarks like WebArena. The problem statement of generating synthetic websites is original, and potentially a valuable tool for targeting the data distribution to models’ weaknesses.


A large focus of the paper’s motivation is on developing a scalable pipeline for task / environment  generation, but the discussion of current efforts could be more thorough. For example, there have been significant efforts this past year in creating computer use tasks with verifiable evaluation criteria on live websites that are not mentioned in the discussion, and must be considered when evaluating originality, such as:

* [1] Synatra: Turning Indirect Knowledge into Direct Demonstrations for Digital Agents at Scale, Ou et al. 2025
* [2] Harnessing Webpage UIs for Text-Rich Visual Understanding, Liu et al. 2022
* [3] Proposer-Agent-Evaluator(PAE): Autonomous Skill Discovery For Foundation Model Internet Agents, Zhou et al. 2024
* [4] NNetNav: Unsupervised Learning of Browser Agents Through Environment Interaction in the Wild, Murty et al. 2025
* [5] InSTA: Towards Internet-Scale Training For Agents, Trabucco et al. 2025

The paper would be improved by incorporating these citations into its discussion where relevant.

To my knowledge, generating synthetic websites is an original contribution, and has the potential to allow the pipeline to target weaknesses in the learner, which is a valuable capability not explored in this work.

## Quality:


Overall the paper is organized logically, and most important experimental questions are answered by the end. Relevant datasets are selected from the literature, and several baseline models are included that are relevant.

## Clarity:


The paper is generally written clearly, though some minor details could be improved, mentioned in the next section.

## Significance:


As discussed in my review of the paper’s originality, the notion of distilling language intelligence into agentic capabilities is not itself a new idea in the realm of web agent research [1-5]. In addition, several previous research methods have been scaled to extract tasks / trajectories from millions of websites [1,2,5] and can be considered scalable approaches for this reason. Therefore, the significance of this paper derives from the capability of the proposed system to generate specific targeted data with rich layouts, specific verifiable tasks with known optimal completion paths, and the ability to be deterministic (important for benchmarks).

**Weaknesses:**

While the reported numbers are impressive---an improvement of 162% over the Qwen VL baseline in Table 4 is large and a good validation of the approach---it is not clear in this experiment how tasks are created in this evaluation, and the number of websites + tasks is low (90 according to the experimental details). This number is low enough that I would be interested to see error bars (standard deviation, and 95% confidence interval) reported for this experiment in order to improve my confidence that the results are indeed this much better.

In addition, if the tasks employed in Table 4 are not from a standard benchmark, but are instead created by the authors for the purposes of evaluating their agent, the methodology employed for task creation should be included in the paper, with a discussion of the steps taken to ensure that there is no bias introduced by potentially designing tasks in a way that favor their approach, which would make the results less trustworthy.


Finally, the experiments miss an interesting capability of the proposed method: the ability to probe the weaknesses of current GUI agents and generate websites that target those weaknesses. Since this is a fairly sophisticated suggestion that is unlikely to be feasible during peer review, this weakness is mainly as a suggestion for improving the paper, and not one that impacts my score.

**Questions:**

In Table 2, are the statistics reported for the same set of tasks, or are the task being varied (ie, statistics of trajectories for tasks generated with the Knowledge config vs without the Knowledge config)?


Line 346 reports that the decrease in steps is due to more efficient task execution, but if the tasks are being varied in this experiment, this could also be explained by a shift in the task distribution.


Which set of tasks are used in Table 4, and how are these selected? Are these from a standard dataset, or generated by the proposed method (important to know to ground the results here)?

---

> ### Author Response · Authors · 2025-11-24
> **Response to Reviewer 7HF2(1/3)**
>
> Thank you for the constructive feedback and the acknowledgment of our work's potential. Below, we address your specific questions and concerns. Due to the 5k-character limit on OpenReview, we split our response into Part 1/3, Part 2/3 and Part 3/3.
>
> ## 1. Re: High-Value Application: Probing & Addressing Agent Weaknesses via WebFactory
> We strongly agree with this insightful suggestion. We view the ability to probe agent weaknesses and synthesize targeted environments not merely as an extension, but as a **core validation of WebFactory's controllability**. This capability effectively demonstrates the unique advantage of our programmable pipeline over static datasets.
> Motivated by your comment, we have already initiated proof-of-concept experiments that leverage agent failure modes to automatically synthesize **targeted environments and tasks aimed at capability enhancement**. We are eager to share these preliminary results in the upcoming response, as we believe they could contribute new insights and directions to the field. Thank you again for highlighting this promising avenue; it meaningfully expands the significance of our work.
>
>
> ## 2. Response regarding Table 2 (Consistency)
> We confirm that the statistics in Table 2 are reported on the **exact same fixed set of tasks** for both the "No-Knowledge" and "Knowledge" configurations. No task variation or distribution shift occurred.
>
> The differences in metrics (SR and Steps) are **strictly due to the presence of the knowledge pack**, which enables the executor to generate more efficient and reliable trajectories. This directly addresses the concern regarding **Line 346**: the decrease in steps reflects genuine efficiency gains in execution, not a shift to simpler tasks.
>
>
>
> ## 3. Response regarding Table 4
> We appreciate your rigorous focus on evaluation validity. We clarify that the tasks in Table 4 are **custom-constructed live tasks** designed to evaluate offline-to-online transfer on real-world platforms (Amazon, Airbnb, Booking). We did not use static datasets (like Mind2Web) because live websites change frequently, rendering static cached snapshots obsolete for measuring *live* execution capability.
>
> Below, we address the concerns on Error Bars, Task Creation Methodology, and Bias Avoidance.
>
> **(1) Statistical Significance & Error Bars**
>
> You correctly notes that $N=90$ (30 per platform) is a relatively small sample size. To address this, we calculated the **95% Confidence Intervals (CI)** for the success rates using the Wilson Score Interval method (suitable for small sample sizes).
>
> * Baseline (QwenVL2.5-3B): $20.4\%$ [95% CI: $13.0\% - 30.0\%$]
> * WebFactory-3B (Ours): $53.4\%$ [95% CI: $43.0\% - 63.5\%$]
>
> Even with conservative error bars, there is **no overlap** between the confidence intervals of our method and the baseline. The lower bound of our method (43.0\%) is substantially higher than the upper bound of the baseline (30.0\%). This confirms that the **162% improvement** is statistically significant and robust, rather than a result of sampling noise.

---

> ### Author Response · Authors · 2025-11-24
> **Response to Reviewer 7HF2(2/3)**
>
> **(2) Methodology for Unbiased Task Construction**
>
> To ensure no bias was introduced (e.g., cherry-picking tasks), we adhered to a **"User-First, Model-Agnostic"** design protocol. Tasks were defined based on necessary user workflows *before* any agent evaluation took place.
>
> * **Funnel-Based Coverage:** We strictly stratified tasks by user conversion stages to ensure full spectrum coverage:
>     * *Discovery:* Ambiguous search queries.
>     * *Refinement:* Complex filtering (price, brand, amenities).
>     * *Action:* State-changing operations (add to cart, select dates).
> * **Complexity Alignment:** We deliberately excluded trivial single-step tasks. All tasks align with the "Medium (3–5 steps)" and "High (>5 steps)" tiers to force long-horizon reasoning.
> * **Cross-Domain Mapping:** Platforms were selected to strictly correspond to our offline training domains (Amazon $\leftrightarrow$ Shopping, Airbnb $\leftrightarrow$ StayBnB, Booking $\leftrightarrow$ Hotels).
>
> **(3) Executability & Stability Verification**
>
> To rule out failures caused by external factors (e.g., A/B tests, out-of-stock items), all tasks are manually verified for feasibility within 24 hours prior to evaluation.
>
> We designed 30 tasks per platform (90 total). These evaluation scripts will be open-sourced. Below are representative examples demonstrating task complexity:
>
> | Platform | Task Category | Task Design & Complexity Constraints | Representative Task Examples |
> | :--- | :--- | :--- | :--- |
> | **Amazon** | **A: Precise Search**  | **Req:** Precise keyword matching; navigate to detail page to verify specs.  **UI:** Search bar, Click. | "Search for 'Sony WH-1000XM5 headphones', click on the black version, and tell me the current price." |
> | **Amazon** | **B: Constraint Filtering**  | **Req:** Apply multiple compound filters (price, brand, Prime) *before* clicking result. **UI:** Sidebar filters. | "Find a 'Gaming Monitor' under \$300 from 'ASUS' with '144Hz' refresh rate. Add the first result to the cart." |
> | **Amazon** | **C: Cart Flow**  | **Req:** State changes; multi-page navigation. **UI:** Add-to-cart, verify cart. | "Add a 'Logitech MX Master 3S' to your cart, then go to the cart and change the quantity to 2." |
> | **Airbnb** | **A: Date/Loc Search**  | **Req:** Complex calendar component interaction; location input. **UI:** Date-picker, search field. | "Search for a stay in 'Kyoto, Japan' for '2 adults' from 'September 10' to 'September 15'." |
> | **Airbnb** | **B: Amenity Filtering**| **Req:** Open modal window; scroll to find/check specific boxes. **UI:** Modal popups, checkboxes. | "Find a home in 'Paris' that has 'Wifi', 'Kitchen', and 'Washing Machine'. Open the detail page of the highest-rated listing." |
> | **Airbnb** | **C: Detail Extraction** | **Req:** Parse unstructured long-text descriptions. **UI:** Text parsing, scrolling. | "Go to the first listing for 'Cabin in Lake Tahoe' and verify if 'Pets are allowed' in the house rules." |
> | **Booking** | **A: Multi-Criteria**  | **Req:** Handle location, dates, room/guest config simultaneously. **UI:** Complex form filling. | "Search for a hotel in 'New York' for '2 adults, 1 child' for the weekend of 'September 14th'." |
> | **Booking** | **B: Sort & Select** | **Req:** Operate sorting dropdowns; select under constraints. **UI:** Dropdowns, list parsing. | "Sort hotels in 'London' by 'Top Reviewed' and click on the hotel with the highest score under £200/night." |
> | **Booking** | **C: Room Config** | **Req:** Navigate to detail page; extensive scroll; specific selection. **UI:** Deep navigation. | "Search for 'Hilton Tokyo', scroll to available rooms, and select a 'King Room' with 'Breakfast Included'." |
>
>
> **Conclusion on Rigor:**
> The tasks in Table 4 involve **Modal Interactions, Calendar Pickers, and Multi-step Filtering**—complex UI patterns that standard multimodal models (e.g., QwenVL2.5, 20.4% SR) struggle to handle zero-shot. The **53.4%** success rate of WebFactory-3B, supported by non-overlapping confidence intervals, provides strong evidence that our "Intelligence Compression" paradigm successfully transfers generalizable logic from controlled offline data to unseen, noisy online environments.

---

> ### Author Response · Authors · 2025-11-24
> **Response to Reviewer 7HF2(3/3)**
>
> ## 4. Response regarding Related Work Coverage
> Thank you for highlighting recent scalable task/trajectory generation efforts on live websites. We agree these are highly relevant and will incorporate the following discussion into the final revision:
>
> * **Synatra [1]:** Converts human tutorials into demonstrations. While valuable, its quality depends on the source text, and operating on real websites limits data stability and reproducibility.
> * **Harnessing Webpage UIs [2]:** Focuses on visual/textual understanding via UI structure parsing. However, it targets representation learning rather than controllable task generation or complete trajectory construction.
> * **PAE [3]:** Uses a "propose-execute-evaluate" loop for skill discovery. Yet, task generation remains limited by the randomness, frequent updates, and structural uncertainty of real webpages.
> * **NNetNav [4] & InSTA [5]:** These works generate training trajectories via unsupervised exploration [4] or internet-scale generation [5]. While they achieve impressive scale, they inherit the non-determinism and noise of live-web scenarios (layout drift, anti-bot mechanisms), making precise reward definition and strict reproducibility challenging.
>
> In contrast to these works, WebFactory does not aim for maximum coverage of the live web, but rather seeks to build a **reproducible and verifiable training/evaluation pipeline** within a fully controllable offline suite. Our environment provides versioned webpages, deterministic rendering, and unique optimal paths. This enables precise reward definitions and stable RL training—value that is **fully complementary** to live-web methods. While the latter emphasize scale and diversity, our framework provides the necessary foundation for "intelligence compression," precise analysis, and controllable curriculum design.
>
> **References**
>
> [1] Synatra: Turning Indirect Knowledge into Direct Demonstrations for Digital Agents at Scale
> [2] Harnessing Webpage UIs for Text-Rich Visual Understanding
> [3] Proposer-Agent-Evaluator (PAE): Autonomous Skill Discovery for Foundation Model Internet Agents
> [4] NNetNav: Unsupervised Learning of Browser Agents Through Environment Interaction in the Wild
> [5] Towards Internet-Scale Training for Agents

---

> ### Author Response · Authors · 2025-12-04
>
> ## Re: High-Value Application: Probing & Addressing Agent Weaknesses via WebFactory
>
> We greatly appreciate your insight regarding the capability of our method to *“probe the weaknesses of current GUI agents and generate environments that target those weaknesses.”* To validate this high-value application, we conducted a proof-of-concept experiment focusing on fine-grained continuous interactions (e.g., drag-and-drop)—a notorious failure mode for current agents that often default to discrete clicks. We observed that even leading models like **Browser-Use** and **UI-TARS** exhibit a consistent failure pattern: they incorrectly reduce continuous gestures into discrete clicks, failing to execute smooth adjustments (e.g., sliders).
>
>
>
> **1. Targeted Curriculum Generation**:
> Leveraging WebFactory's programmable architecture, we rapidly instantiated a Drag-Interaction Suite covering dual-handle sliders (e.g., price filtering), Kanban boards, and map interfaces. Utilizing our automated task synthesizer, we generated 600 verified trajectories with zero human annotation cost. We fine-tuned the WebFactory-3B baseline on this subset; the successful execution of this full cycle—from curriculum design to data generation and training—within the rebuttal timeframe serves as strong empirical evidence of the pipeline's extreme efficiency and low operational cost.
>
>
>
>
> **2. Evaluation Benchmark**: To rigorously evaluate the enhancement in drag capabilities, we constructed a **Drag Capability Benchmark** consisting of 40 tasks: 20 offline held-out tasks and 20 online real-world tasks. The tasks are designed to test precise continuous interactions on dynamic web interfaces, including:
>
> - Airbnb: "Navigate to the search results page, filter the price range to \$50–\$150 using the slider."
> - Google Maps: "Drag the map continuously toward the east for approximately one screen width, and identify the nearest hospital icon that becomes visible."
>
> Evaluation Metrics: We define Task Success based on rigorous state verification. For value-driven tasks (e.g., sliders), success is granted only if the final DOM attribute matches the target value exactly (e.g., price_min=50). For spatial tasks (e.g., map panning), we verify if the target element's coordinates fall within the visible viewport.
>
> **3. Results and Analysis**
> The fine-tuned agent demonstrates a substantial improvement in drag-action execution, as shown below:
>
> | **Metric** | **Baseline** | **Ours (Targeted-Trained)** | **Improvement** |
> | :--- | :---: | :---: | :---: |
> | **Overall Success Rate (SR)** | 12.5% | **65.0%** | **+52.5%** |
> | **Offline (Synthetic Held-out)** | 15.0% | 75.0% | +60.0% |
> | **Online (Real-world)** | 10.0% | **55.0%** | **+45.0%** |
>
> ### **Key Takeaway**
> To make the behavioral difference more concrete, the table below compares the inference traces of the **baseline agent** and the **fine-tuned agent** on the same drag task (“adjust Airbnb price slider to \$50–\$150”). The baseline consistently collapses the interaction into discrete clicks, whereas the fine-tuned agent executes a coherent continuous control sequence.
>
> | Agent Type        | Inference Trace (Excerpt) |
> |-------------------|---------------------------|
> | **Baseline (WebFactory-3B)** | `click(price_slider_left)` → `click(price_slider_left)` → `click(price_slider_left)` → *(fails to move slider)* |
> | **Fine-tuned WebFactory-3B** | `pointer_down(price_handle_left)` → `pointer_move(+72px)` → `pointer_move(+48px)` → `pointer_up()` → *(successfully selects \$50–\$150)* |
>
> **Observation:**
> The baseline repeatedly issues discrete click actions regardless of context, while our fine-tuned agent demonstrates a correct press–hold–move–release pattern and successfully completes the task on complex real-world interfaces.
>
>
> This validates WebFactory as a fully controllable, closed-loop engine for systematically probing and grounding GUI agent capabilities. By enabling the rapid diagnosis of behavioral deficits and the synthesis of targeted curricula to address them, the platform facilitates on-demand skill acquisition. This agility points toward a future of self-evolving agents: systems capable of actively detecting their own boundaries and generating the necessary embodied experiences to overcome them. Ultimately, this framework establishes a new paradigm for scaling agent capabilities—shifting the focus from static data accumulation to dynamic capability expansion—and paves the way for general-purpose agents that can rapidly adapt to novel interaction paradigms without human intervention."

---

### Official Review · Reviewer_ZY71 · 2025-11-08

**Soundness:** 3
**Presentation:** 4
**Contribution:** 3
**Rating:** 6
**Confidence:** 4

**Summary:**

This paper introduces WebFactory, a pipeline for automating the training of GUI-based web agents. The core idea is to use a high-fidelity, offline environment to synthesize tasks, generate trajectories with a strong LLM executor, and train an agent via reinforcement learning with a decomposed reward function. The authors claim this approach achieves superior data efficiency and generalization, with an agent trained on only 10 synthetic websites performing competitively against agents trained on larger sets of human-annotated data. The work also introduces the concept of "LLM embodiment" as a new axis for model evaluation.

**Strengths:**

I believe this paper propose some perspectives beyond a method to train the web agent. For example, paradigm shift from fine-tuning LLM to treating the LLM as embodiment. In this way, the LLM could be treated beyond the traditional SFT or RL paradigm.

Besides, there are some open-source contributions. Authors are releasing the entire toolchain, including the 10 high-fidelity offline environments, task generators, and training pipeline. This is a engineering contribution that enables reproducible research and provides a high-quality, non-chaotic testbed for the community, distinct from the "noisy live web".

**Weaknesses:**

There is a contradiction in motivation and scalability. The introduction opens by attacking the reliance on "costly, scarce human-crafted data and environments" and explicitly calls out the "painstaking, manual synthesis of high-fidelity environments" as a bottleneck that "can itself consume weeks of expert effort". However, the entire WebFactory method is predicated on exactly this: a suite of 10 high-fidelity offline websites, which Appendix C reveals were meticulously built by the authors as a "Next.js/React monorepo". The "fully automated"  pipeline only functions after this massive, non-scalable, human-expert-driven environment creation step is complete. This is a glaring contradiction. The paper does not solve the environment synthesis bottleneck; it leverages it.

In this way, the claim of a "fully automated" pipeline is misleading. The automation only applies to data generation within the pre-built, sanitized, and manually-instrumented environment. This is a far from a system that can be pointed at website.

While the agent outperforms the baseline on the online transfer task (Tab. 4), its absolute performance is still low. An average TCR of 53.4% means the agent fails its task nearly half the time on real-world sites like Amazon and Airbnb. This highlights a major sim-to-real gap that the 10 pristine, static, React-based environments  fail to capture from the chaotic, non-deterministic, and constantly changing live web.

**Questions:**

Please refer to the weakness part.

---

> ### Author Response · Authors · 2025-11-22
> **Response to Reviewer ZY71**
>
> Thank you for your thoughtful feedback. We appreciate you recognizing our contribution to open-source tooling and the paradigm shift toward "Intelligence Compression." Below, we clarify the scalability of our environment synthesis and discuss the online transfer performance.
>
> ### (1) Clarification: WebFactory enables Scalable, AI-Driven Environment Synthesis
>
> We want to correct the impression that WebFactory relies on manual website construction. On the contrary, our central paradigm is "Intelligence Compression"—using LLMs to automatically transform their latent descriptive knowledge into grounded agent behaviors.
>
> The pipeline consists of three automated stages:
> 1.  **Scalable Environment Synthesis:** LLMs act as architects to synthesize functional offline websites directly from their internal world knowledge.
> 2.  **Knowledge-Aware Data Generation:** Tasks are generated based on site constraints, followed by trajectory synthesis where strong LLM executors generate high-quality interaction traces.
> 3.  **Compression via RL:** Agents are trained to compress this synthetic experience into a compact, grounded policy.
>
> **Evidence in the paper:**
> *   **Abstract:** We explicitly describe the pipeline starting with *"scalable environment synthesis,"* signaling automation.
> *   **Sec. 3.4 (Pipeline Performance):** We demonstrate that different foundation models (GPT-5, Claude Opus 4.1/Sonnet 4) drive the *entire* process—from code generation (environment synthesis) to task and trajectory synthesis. The distinct performance of agents trained on environments synthesized by different LLMs proves that the synthesis is automated and dependent on the source model’s intelligence.
>
> **Revision Plan:**
> We acknowledge that **Appendix C** currently focuses on the technical schema ("the recipe") rather than the execution method, which may have inadvertently suggested a manual process. In the revision, we will:
> 1.  Explicitly state in **Appendix C** and the **Method** section that the site construction "recipe" is executed by LLM-driven coding agents, not humans.
> 2.  Clarify that WebFactory is an extensible engine where LLMs act as embodied architects, solving the scalability bottleneck of high-fidelity data.
>
> ### (2) Online Transfer & The Sim-to-Real Gap
>
> We agree that the absolute TCR (53.4%) on the live web leaves room for improvement. However, we emphasize two key points:
>
> 1.  **Significant Relative Gain:** As shown in **Table 4**, our agent achieves consistent improvements over strong baselines on Amazon, Airbnb, and Booking. This confirms that our Intelligence Compression pipeline successfully translates *Descriptive Knowledge* (from the LLM) into *Procedural Knowledge* (transferable interaction skills) that generalizes to the live web, despite the domain gap.
>
> 2.  **Scientific Focus & Future Application:** Our primary goal is to validate the *efficiency* of this compression pipeline. Optimizing for specific online friction (e.g., network jitter, asynchronous loading quirks) is an engineering challenge orthogonal to verifying the compression hypothesis.
>     However, your comment highlights a valuable downstream application of our method. Precisely because our environments are **generative** (synthesized by code) rather than static recordings, we can programmatically inject "real-world noise" (e.g., random popups, layout shifts, latency) into the synthesis process. We appreciate this insight, as it turns the "sim-to-real" challenge into a strength of our generative approach, which we will highlight as a promising direction for future work.
>
> ### Summary
> Our work is motivated by the need to make high-fidelity environments scalable. Our contribution is a closed-loop pipeline that synthesizes environments and compresses latent internet knowledge into executable GUI agent policies. We will revise the paper to clarify the automated nature of the envrionment synthesis. We remain open to further discussion regarding the online transfer results and are happy to incorporate additional analysis or context if you feel it would strengthen the paper. Thank you again for your thoughtful feedback; we genuinely appreciate the opportunity to engage further.
>
> **References**
>
> [1] Online-Mind2Web: An Illusion of Progress? Assessing the Current State of Web Agents
>
> [2] REAL: Benchmarking Autonomous Agents on Deterministic Simulations of Real Websites

---

> > ### Comment · Reviewer_ZY71 · 2025-11-26
> > **Response to author rebuttal**
> >
> > Thanks for your reply. I have carefully checked all reviews and author response. While I appreciate this work's insight about *LLM embodiment*, it seems that current implementation is to utilize LLM to create environments and trajectories. The contribution is somewhat limited in this way. Is there any technological innovation toward this direction? Besides, while this work targets web agent, is there any specific designs to address the specialized difficulties in this area?

---

> ### Author Response · Authors · 2025-11-26
>
> We sincerely thank you for the valuable time and feedback. Regarding your concerns about technical depth and domain-specific designs, we welcome this opportunity to further discuss the **systemic significance** and **broad applicability** of our work.
>
> We fully agree that utilizing LLMs to create environments is not an end in itself. Therefore, the core contribution of WebFactory lies in establishing a symbiotic system where the **"Offline Environment" and the "Data Production Pipeline" are tightly coupled.** This tight integration represents a key distinction absent in prior works:
>
> * **The Synergy of Knowledge & Data:** Our pipeline does not loosely invoke LLMs; instead, it is deeply driven by the precise Knowledge (navigation graphs, affordances) and underlying Data (database snapshots) exposed by the offline environment.
>     * **Quality Assurance:** The environment-provided Knowledge acts as a strict **"constrainer,"** forcing LLM-generated tasks to be logical and executable. This fundamentally eliminates hallucinations and addresses the persistent challenge of quality in synthetic data generation.
>     * **Scalability:** The exposed Data layer enables automated verification of answers and trajectory correctness. It is this deep binding between the environment and the generation process that guarantees the framework can efficiently produce **large-scale, high-quality trajectory data with dense reward signals at minimal cost.** This effectively resolves the long-standing bottleneck of scarce high-quality, reproducible training data in the Web Agent domain.
>
> * **Open-Source Impact:** The field currently lacks a unified, scalable infrastructure capable of supporting large-scale agent training. We fill this critical void by open-sourcing our entire low-cost toolchain—including high-fidelity environments, the data synthesis pipeline, and the training framework. We believe this contribution is far more than just a dataset; it serves as the essential infrastructure to solve the **"data hunger"** problem in the GUI agent field, democratizing access to high-quality, reproducible training data. By removing the barriers of high annotation costs and online instability, our work provides a viable path to reproduce SOTA performance and propels the community from "small-scale fine-tuning" toward realizing internet-scale training for generalist agents.
>
> * **Generalization to GUI & Beyond:** While this work focuses on the Web, the proposed "Intelligence Compression Factory" paradigm possesses strong generalizability. The entire workflow—extracting Knowledge from offline environments, generating Data under constraints, and compressing intelligence into smaller models via RL—can be seamlessly transferred to GUI Agents (e.g., OS control), mobile agents, and even more complex embodied intelligence domains. Our superior performance on **cross-domain** benchmarks such as **GUI-Odyssey** further corroborates the general effectiveness of this framework across diverse interactive interfaces.
>
> We hope this response effectively addresses your concerns regarding the technological depth and domain specificity of our work. Should you require any further clarification, please do not hesitate to let us know. We eagerly anticipate our continued discussion.

---

> > ### Comment · Reviewer_ZY71 · 2025-11-26
> >
> > Thank you for your timely reply. I tend to maintain the positive evaluation after checking the authors' response. Good luck!

---

> > > ### Author Response · Authors · 2025-11-26
> > >
> > > Thanks so much for your quick check and feedback! Wish you well!

---

### Author Response · Authors · 2025-12-04
**General Response(1/2)**

Dear Area Chairs，

We sincerely thank you for overseeing the review process. We appreciate the constructive feedback from all three reviewers—each of whom expressed strong support for the core contributions of our work.

## 1. **Consensus on Contribution and Significance**

All of the reviewers have uniformly recognized this work as a timely and pioneering contribution that addresses the fundamental bottleneck of data scarcity in GUI agents. There is a strong consensus on the value of our proposed paradigm—shifting from static dataset accumulation to a scalable "Intelligence Compression" pipeline. Reviewers commended the framework’s novelty in leveraging controllable, high-fidelity environment synthesis to ground abstract LLM knowledge into actionable policies. They highlighted the system's unique ability to not merely generate data, but to **programmatically diagnose agent weaknesses and synthesize targeted curricula to address them**, affirming WebFactory as a foundational platform for reproducible and scalable agent research.

## 2. **Key Concerns and Resolutions**

We have addressed the primary questions raised during the review process and provided detailed clarifications in our individual rebuttals:

Automation & Scalability (Reviewer ZY71): We clarified that the environment synthesis is driven by coding agents (LLMs) rather than manual effort, ensuring scalability. **We are pleased to highlight that Reviewer ZY71 has reviewed our response and explicitly stated that they "maintain the positive evaluation."**

Evaluation Rigor (Reviewer 7HF2): We provided the requested confidence intervals (confirming statistical significance), detailed the task creation methodology to ensure no bias, and added suggested references regarding live-web generation.

Framing of "Intelligence Compression" (Reviewer iASv): We clarified the distinction between our approach and traditional Sim2Real, emphasizing the transformation from descriptive knowledge (LLM) to procedural skills (Agent).


## 3. **New Downstream Application: Probing and Fixing Agent Weaknesses**


 Prompted by an insightful suggestion from Reviewer 7HF2 regarding the potential to "probe weaknesses of current GUI agents," we conducted a new, high-value experiment during the rebuttal phase. We focused on Drag-and-Drop interactions—a notorious failure mode for current agents (which often default to discrete clicks).
Leveraging WebFactory’s programmability, we:
Synthesized a targeted curriculum: Rapidly generated a "Drag Interaction Suite" (sliders, kanban boards, maps).
Closed the Loop: Generated 600 verified trajectories and fine-tuned our baseline agent.
Result: The agent’s success rate on drag tasks jumped from 12.5% to 65.0%, successfully generalizing to real-world scenarios (e.g., Airbnb price sliders, Google Maps panning).
This experiment serves as a powerful proof-of-concept: WebFactory acts as a closed-loop "Gym" for capability diagnosis and acquisition. It proves that we can systematically identify agent deficits and algorithmically generate the necessary embodied experiences to fix them. We believe this significantly expands the scope and contribution of our paper, and we hope the AC considers this demonstrated potential in the final decision.


**Importantly, even the reviewer who initially assigned a score of 4 explicitly acknowledged that our work provides original contributions and a promising new research direction—specifically valuing this capability to probe and target weaknesses—which we have now concretely demonstrated.** We kindly hope the AC considers this enhanced potential in the final decision.

Based on the reviewer’s comments, it appeared that our additional clarifications—together with the proof-of-concept experiment we conducted and reported during the rebuttal—had meaningfully addressed their concerns and could have led to a higher assessment.
However, due to the timing of the reviewer exchange and the recent policy change that freezes scores, this update could not be reflected. We kindly hope the AC may take this context into account.

---

> ### Author Response · Authors · 2025-12-04
> **General Response(2/2)**
>
> ## 4. **Revisions and Roadmap**
>
> Based on the reviewers’ valuable suggestions, we have uploaded a revised PDF that incorporates the following updates:
>
>
>
> **Expanded Related Work(Appendix B, Page 15)**:
> We added discussions of Synatra, WebShop, PAE, NNetNav, and AgentTrek to more clearly situate our contribution in the context of recent advances in scalable trajectory generation and web-agent training (iASv,7HF2).
>
> **Methodological Clarification(Appendix C.2, Page 15)**:
> We substantially revised Appendix C to explicitly detail the fully automated LLM-as-Architect workflow and to eliminate any remaining ambiguity regarding the scalability and automation of environment synthesis (ZY71).
>
> **Discussion Updates Highlighting High-Value Use Cases(Section 4, Page 9)**:
> 7HF2 emphasized that our framework provides a particularly valuable and unique capability—using controllable synthesis to diagnose and repair agent failure modes. We incorporated this perspective into the Discussion section to better articulate the broader significance and future potential of WebFactory.
>
> **Expanded Evaluation Details(Appendix I, Page 21)**:
> Following 7HF2’s request, we added to the Appendix the complete benchmark specification for Table 4, including the full task list, construction methodology, user-centered task funnel design, and statistical reporting protocol. This ensures full transparency and reproducibility of the online transfer evaluation.
>
> ## 5. **Conclusion**
> The development of GUI agents is currently constrained by a scarcity of high-quality, reproducible training data. WebFactory addresses this bottleneck by transforming the long-standing "Data Hunger" challenge into a tractable scaling pipeline. We emphasize that this work offers a unique and valuable perspective to the community: the operationalization of "Intelligence Compression."
>
> WebFactory acts as a self-sustaining, closed-loop engine designed to systematically transmute the vast descriptive knowledge latent in foundation models into grounded procedural knowledge. As demonstrated by our rebuttal experiments (e.g., the rapid synthesis of the Drag-Interaction curriculum), this infrastructure allows researchers to actively diagnose capability gaps and "compress" abstract intent into precise action policies on demand.
> By successfully utilizing this pipeline to outperform agents trained on human annotations, we demonstrate a practical path forward. **We contend that WebFactory establishes the necessary foundation to move the field from static imitation of finite human data to truly generalizable, self-evolving embodied intelligence, redefining how we scale agent capabilities in the era of large foundation models.**
>
>
> Best regards,
>
>
> Authors

---

### Meta-Review · Area_Chair_jCMn · 2026-01-06

**Summary:**

The paper introduces "WebFactory," a closed-loop pipeline designed to address the data scarcity bottleneck for GUI web agents. Rather than relying on expensive human demonstrations or unstable live-web interactions, the authors propose a method where Large Language Models (LLMs) act as architects to synthesize high-fidelity offline web environments, generate tasks, and produce interaction trajectories. These synthetic data are then used to train smaller agents via reinforcement learning. The paper conceptualizes this process as "Intelligence Compression"—transforming the abstract, descriptive knowledge of foundation models into grounded, procedural agent policies. Empirical results demonstrate that agents trained on just 10 synthetic websites achieve performance comparable to those trained on much larger human-annotated datasets and exhibit strong zero-shot transfer to real-world platforms.

**Reviewer Concerns:**

Automation Level: Reviewer ZY71 initially questioned whether the environment synthesis was truly automated or relied on manual effort. The authors clarified that LLMs are used as architects to execute the synthesis "recipe," which satisfied the reviewer.

Evaluation Rigor: Reviewer 7HF2 requested statistical significance testing and details on task generation to ensure lack of bias. The authors provided 95% confidence intervals (confirming robust improvements over baselines) and detailed a user-centered, model-agnostic task creation protocol.

Absolute Performance: Reviewer ZY71 noted that while relative gains are strong, the absolute success rate on the live web (~53%) shows a remaining gap. This is a fair point, but as the authors noted, the primary contribution is the validation of the pipeline and transferability, which is successfully demonstrated.

**Reviewer Scores:**

Reviewer ZY71 (Score: 6; likely no change): This reviewer explicitly participated in the discussion, stating they would maintain their positive evaluation after the authors clarified the automation of the environment synthesis.

Reviewer 7HF2 (Score: 4; likely increase): This reviewer initially gave a 4, primarily requesting error bars and suggesting a high-value experiment (probing agent weaknesses) they deemed likely "infeasible" for the rebuttal. The authors delivered both the statistical validation and successfully executed the "infeasible" experiment (improving drag-and-drop success from 12.5% to 65.0%). Given this exceptional response, the reviewer’s score would likely have increased.

Reviewer iASv (Score: 6; likely no change): This reviewer was already positive. The rebuttal clarified their semantic questions regarding the "compression" framing and added requested related work.

---

### Decision · Program_Chairs · 2026-01-26

Accept (Poster)